# A STING-based biosensor affords broad cyclic dinucleotide detection within single living eukaryotic cells

Alex J. Pollock[1,2], Shivam A. Zaver[1,2] & Joshua J. Woodward [1✉]

Cyclic dinucleotides (CDNs) are second messengers conserved across all three domains of life. Within eukaryotes they mediate protective roles in innate immunity against malignant, viral, and bacterial disease, and exert pathological effects in autoimmune disorders. Despite their ubiquitous role in diverse biological contexts, CDN detection methods are limited. Here, using structure guided design of the murine STING CDN binding domain, we engineer a Förster resonance energy transfer (FRET) based biosensor deemed BioSTING. Recombinant BioSTING affords real-time detection of CDN synthase activity and inhibition. Expression of BioSTING in live human cells allows quantification of localized bacterial and eukaryotic CDN levels in single cells with low nanomolar sensitivity. These findings establish BioSTING as a powerful kinetic in vitro platform amenable to high throughput screens and as a broadly applicable cellular tool to interrogate the temporal and spatial dynamics of CDN signaling in a variety of infectious, malignant, and autoimmune contexts.

[1] Department of Microbiology, University of Washington, Seattle, WA 98195, USA. [2]These authors contributed equally: Alex J. Pollock, Shivam A. Zaver.
✉email: jjwoodwa@uw.edu

The mammalian innate immune system provides a critical first line of defense against invading microorganisms through a suite of germline encoded, invariant sentinel proteins called Pattern Recognition Receptors (PRRs). PRRs survey the extracellular and intracellular milieu for molecular signatures of microbial origin, termed pathogen associated molecular patterns (PAMPs)[1,2]. Upon engaging PAMPs, PRRs participate in signal transduction cascades to activate anti-microbial gene regulatory programs that ultimately facilitate pathogen clearance[1,2]. Microbial nucleic acids, including DNA and RNA species, constitute a major class of PAMPs. Although a number of nucleic acid sensing PRRs have been identified to-date, the enzyme cyclic GMP-AMP Synthase (cGAS) has emerged as one of the most important sensors of foreign and self double-stranded (ds) DNA[2–5].

Upon allosteric activation by dsDNA, cGAS catalyzes the production of the cyclic dinucleotide (CDN) second messenger 2'3'-cyclic GMP-AMP (c[G(2',5')pA(3',5')p], 2'3'-cGAMP, or cGAMP), which then directly binds to and activates the ER-resident, scaffold protein STING[6–10]. In addition to 2'3'-cGAMP, STING also recognizes 3'3'-linked cyclic dipurines, including c-di-AMP, c-di-GMP, and 3'3'-cGAMP, produced in the context of infection with certain bacterial species, although with markedly reduced affinities[11–14]. CDN-mediated activation of STING facilitates the recruitment and activation of several kinases, culminating in transcription factor-mediated cytokine expression and induction of autophagy to sterilize the cytosol of the infected cell[15–19].

Despite significant advances in our understanding of cGAS and STING regulation, the development of methods for monitoring the kinetics and dynamics of CDN signaling, especially in living cells, is limited. Most studies have relied on type I interferon (IFN-I) induction downstream of STING activation as an indirect reporter of cGAS activity in cells[11,13,20]. While these assays are robust and sensitive, they are not specific as many PAMPs can elicit IFN-I responses. To that end, fluorescent tagged-STING constructs have been used as a more direct reporter for STING activation. These assays rely on the translocation of STING to a perinuclear punctate compartment upon CDN binding as a qualitative proxy for its activation and have recently been employed to monitor 2'3'-cGAMP transfer via gap junctions[21]. These tools, however, are limited to a qualitative, binary localization readout. In lieu of these reporter assays, methods to directly measure cyclic dinucleotides, including mass spectrometry, enzyme immunoassays (EIA), ENPP1-based luciferase assays (cGAMP-luc), and RNA-based biosensors have been developed[3,4,6,13,22–25]. While these assays are specific, they range in their sensitivity and only provide bulk endpoint measurements following destruction of the biological sample.

FRET biosensors have been developed for the detection of small molecules, including the nucleotide second messengers cAMP[26,27], cGMP[28], and cyclic-di-GMP[29–34]. Ligand binding to an intramolecular FRET biosensor results in a conformational shift that alters the relative distance and orientation of fused compatible donor and acceptor fluorophores. This, in turn, alters the excitation energy transfer between fluorophores, which can be quantitated by exciting the donor fluorophore and determining the ratio of acceptor fluorophore emission to donor fluorophore emission. This change in fluorescence is directly linked to ligand occupancy and thus reports on ligand concentration either in solution or within cells[35–37]. Because these biosensors are based on native ligand binding proteins, they are powerful, genetically encodable tools with biologically relevant binding affinities and responses. Use of these biosensors has provided fundamental insight into the spatial and temporal dynamics of nucleotide signaling, as well as the identification of activators and inhibitors following in vitro and cellular screens.

To address the limitations of current CDN detection techniques, we report the development of an intramolecular FRET biosensor deemed BioSTING, which is based on the eukaryotic CDN binding protein STING[38]. Here, we demonstrate that recombinant BioSTING is a sensitive tool capable of detecting real-time CDN synthesis, drug inhibition, and extracted CDNs from cellular sources. Further, we show that BioSTING expressed in eukaryotic cells can be used to detect CDN synthesis, import, localization, and degradation by a viral protein. Thus, BioSTING affords a powerful in vitro and cellular platform for monitoring CDN levels and is likely to facilitate fundamental discoveries relating to CDN biology as well as translational drug discovery campaigns.

## Results

**Design and development of BioSTING.** CDN signal transduction is mediated by nucleotide binding to effector proteins. Ligand induced structural changes in the protein alter effector function to execute changes in response to altered cellular concentrations of the second messenger[39]. The mammalian protein STING is unique in its ability to bind a variety of CDNs of both mammalian and bacterial origin[6–14]. Structurally, STING is a multipass membrane protein with a C-terminal domain consisting of a dimerization region, a CDN binding domain, and a C-terminal tail (CTT) that is required for downstream signaling (Fig. 1a)[15–19]. The CDN binding domain has been extensively characterized at atomic resolution. When the structures of human STING in the apo and c-di-GMP bound states are aligned, one monomer in each dimer was observed to overlap with nearly no change. However, the second monomer exhibited translocation of over 4 and 12 angstroms in the N and C-termini, respectively (Fig. 1b, Supplementary Fig. 1a, b). Because FRET is highly sensitive to the intermolecular distance between compatible fluorophores, we hypothesized that a fluorescent fusion protein with the STING CTD would afford a platform for the development of a FRET-based CDN biosensor (Fig. 1c)[39–41].

Unlike some human STING alleles, murine STING (mSTING) binds both bacterial and eukaryotic dipurine containing CDNs[6–14]. Therefore, to construct a reporter of broad utility, a prototype sensor was generated by fusing mSTING to the bright and photostable FRET pair mTFP and mKO2 at L152 and E335, respectively (Fig. 1a)[35–37,42,43]. The residues contained within this region of mSTING include the dimerization domain and the CDN binding domain but exclude the amino-terminal transmembrane domains and CTT. Based on analysis of the crystal structure, we hypothesized that, upon cyclic dinucleotide binding, the C-terminus fused to mKO2 would shift in closer proximity to mTFP and increase the amount of FRET signal (Fig. 1c). Indeed, we observed a modest ~7% FRET increase in the presence of cGAMP with purified recombinant protein (Supplementary Fig. 1c). While this observation demonstrated the utility of this approach, the FRET signal was deemed insufficient, and we sought to optimize this prototype to enhance the FRET signal. Notably, we did not observe a FRET change using the eCFP and eYFP FRET pair highlighting the importance of fluorophore orientation as a factor in generating a FRET response.

FRET is exquisitely sensitive to changes in distance and orientation. As such, a five amino acid GGSGG linker was added between mSTING-CTD and each fluorophore individually and in combination[44,45]. While addition of the linker between STING-CTD and mKO2 slightly diminished the FRET response, addition of GGSGG between mTFP and STING-CTD alone successfully increased the FRET change to over 20% (Fig. 1d, Supplementary Table 1, and Supplementary Fig. 1c). This modified FRET biosensor responded to 2'3'-cGAMP with a dynamic range of

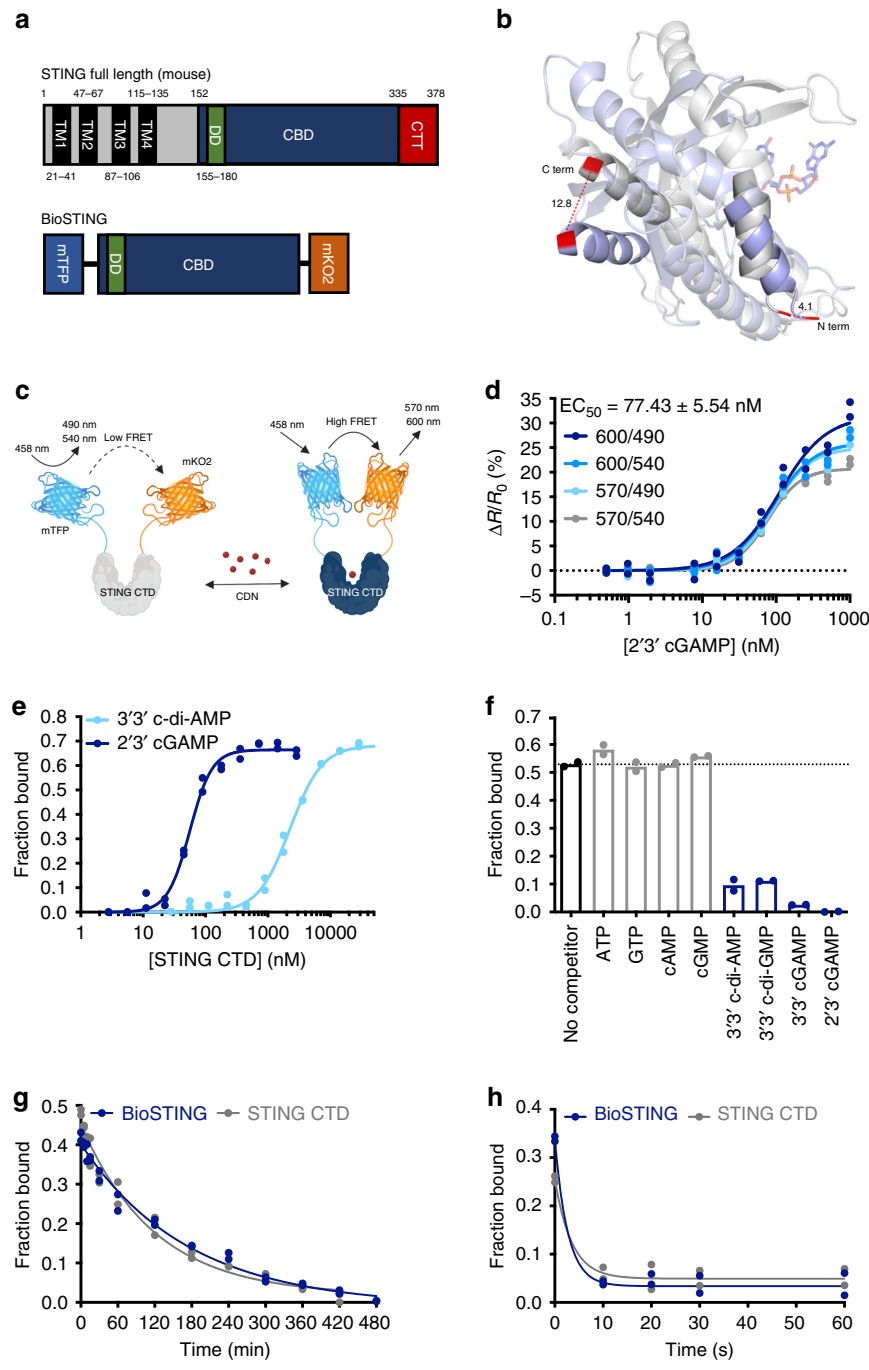

**Fig. 1 BioSTING design and optimization. a** Schematic representation of full-length STING (top) and BioSTING with additional GGSGG linker between mTFP and STING-CTD (bottom). Abbreviations are defined as follows: TM (transmembrane region), DD (dimerization domain), CBD (CDN binding domain), and CTT (C-terminal tail). **b** Overlay and structural alignment of human STING CTD in apo (blue) and c-di-GMP bound (gray) states (PDB 4F5E and 4F5D, respectively). Only a single monomer, which exhibits the largest structural rearrangement upon ligand binding, is shown. The N and C-terminal helices of the CTD are highlighted with the terminal residues in red and their associated displacement following ligand binding labeled in angstroms. **c** Model of the FRET increase which occurs upon CDN binding. Generated with Biorender.com. **d** Recombinant BioSTING FRET response to increasing concentrations of 2'3'-cGAMP using 458 nm excitation and listed emission wavelengths. **e** DRaCALA radioactive nucleotide binding assay of BioSTING using ~1 nM [$^{32}$P] labeled 2'3'-cGAMP and 3'3'-c-di-AMP. Corresponding STING-CTD binding curve is Fig S1D. **f** DRaCALA binding analysis of BioSTING using ~1 nM [$^{32}$P] labeled 2'3'-cGAMP in the presence of excess (500 μM) unlabeled nucleotides. Corresponding STING-CTD competition is Fig S1e. **g** Time course of excess unlabeled 2'3'-cGAMP competing off bound [$^{32}$P] labeled 2'3'-cGAMP from BioSTING (blue) and STING-CTD (gray). **h** Time course of excess unlabeled 3'3'-c-di-AMP competing off bound [$^{32}$P] labeled c-di-AMP from BioSTING (blue) and STING-CTD (gray). In panels **g** and **h**, BioSTING was pre-incubated with ~1 nM of [$^{32}$P] labeled CDNs for 10 min followed by the addition of 1 mM unlabeled 2'3'-cGAMP (**g**) or 3'3'-c-di-AMP (**h**). In all panels, individual data points of $n = 2$ biological replicates are shown.

12–125 nM and a limit of detection (signal-to-noise ratio 3:1) of approximately 12 nM (Supplementary Table 1). Because a 20% FRET increase is more than sufficient for cellular applications, we named this protein BioSTING (Biosensor STING) and used this version in all further experiments.

**Characterization of BioSTING-CDN binding.** Having generated BioSTING, we next sought to compare the biochemical parameters of nucleotide interactions between purified BioSTING and STING-CTD by DRaCALA analysis[46]. Radioactive cGAMP (Supplementary Fig. 1d) bound BioSTING and STING-CTD at a Kd of 56 and 61 nM, respectively (Fig. 1e, Supplementary Fig. 1e). Radioactive c-di-AMP (Supplementary Fig. 1d) bound BioSTING and STING-CTD at a Kd of 2.26 μM and 2.58 μM, respectively (Fig. 1e, Supplementary Fig. 1e). These results confirmed that the region of STING contained within BioSTING maintained CDN binding and that the addition of flanking FRET-fluorophores did not alter binding affinity or disrupt the dimer to which CDNs bind. To confirm that BioSTING retained binding specificity, we performed radioactive cGAMP binding analysis with both BioSTING and STING-CTD in the presence of a variety of unlabeled nucleotides. As expected, only 3′3′-cyclic and 2′3′-cyclic dinucleotides but not ATP, GTP, cAMP, or cGMP could compete off bound [$^{32}$P] 2′3′-cGAMP (Fig. 1f and Supplementary Fig. 1f). These findings reveal that fusion of the two FRET fluorophores to BioSTING has no discernable impact on nucleotide specificity or binding affinity relative to STING-CTD.

Finally, we monitored the dissociation kinetics by determining the rate that cold nucleotide could compete off bound radioactive nucleotide. Radioactive 2′3′-cGAMP was competed off BioSTING and STING-CTD by cold 2′3′-cGAMP with a half-life of 121 and 83 min, respectively (Fig. 1g). Radioactive 3′3′-c-di-AMP was competed off of both proteins with a half-life of <10 s by cold 3′3′-c-di-AMP (Fig. 1h). Together, these results show similar dissociation rates of CDNs from BioSTING relative to STING-CTD, consistent with the similarity in the observed Kd's for these nucleotides. In addition, due to the rapid dissociation rate of 3′3′-c-di-AMP, BioSTING is anticipated to afford rapid monitoring of both increases and decreases in nucleotide levels of 3′3′-CDNs. However, as a consequence of the relatively slow dissociation rate of 2′3′-cGAMP, BioSTING may afford real time monitoring of increases in 2′3′-cGAMP but may be limited in the temporal resolution associated with its decline. Notably, these results suggest that STING activation by 2′3′-cGAMP may be possible through a single binding event while activation by 3′3′-CDNs may require constant exposure to an activating concentration of ligand.

**BioSTING provides a real-time readout of cyclic dinucleotide production in vitro.** Current in vitro methods for monitoring CDN production employ sensitive EIA, mass spectrometry, and cGAMP-luc endpoint measures, as well as less sensitive continuous tools, including a cGAMP RNA biosensor and an indirect pyrophosphate release assay[3,4,6,22–25]. Given the limitations of current CDN detection methods, the ease of recombinant BioSTING production, and its capacity to directly report on a variety of CDNs at biologically relevant concentrations, we employed BioSTING to monitor in vitro CDN synthase activity and inhibition to demonstrate its utility for kinetic characterization and feasibility for high throughput screening related to these enzymes.

To demonstrate the ability of recombinant BioSTING to detect real time production of cyclic dinucleotides, we monitored DNA integrity scanning protein A (DisA) mediated synthesis of 3′3′-c-

di-AMP, which occurs constitutively in the presence of ATP. In reactions consisting of BioSTING, ATP, and increasing concentrations of purified DisA, we observed increasing rates of FRET signal concomitant with increasing DisA concentrations (Fig. 2a). Next, cGAS synthesis of 2′3′-cGAMP from ATP and GTP was monitored. As expected, the rate of BioSTING FRET signal correlated with increased cGAS protein and there was no 2′3′-cGAMP production without the addition of DNA, which is required for allosteric activation of the enzyme (Fig. 2b, Supplementary Fig. 2a).

BioSTING FRET assays are carried out in a 96-well format using a fluorescent plate reader making it amenable to in vitro high throughput screening efforts (Supplementary Fig. 2b). Recently, Pfizer reported the development of the cGAS inhibitor PF-06928215 [52]. To test the use of BioSTING as a platform to characterize small molecule inhibitors, we performed dose response measurements with fixed cGAS and variable concentrations of PF-06928215 (Fig. 2c). In parallel, we also titrated cGAS levels while keeping the concentration of PF-06928215 fixed (Supplementary Fig. 2c). The slope of the linear region was determined for each reaction, plotted versus PF-06928215 concentration, and fit to identify an IC50 of ~12 μM (Fig. 2d), in close agreement with the reported IC50 of 5 μM[47]. Together these studies demonstrate that BioSTING affords robust, continuous detection of CDN levels in vitro and offers flexibility in assay design, making it amenable to high throughput screens and kinetic characterization studies of a variety of CDN synthases. In addition to screening for modulators of CDN synthase activity, BioSTING could also potentially be used to screen directly for STING agonists and antagonists.

**BioSTING can detect 2′3′-cGAMP extracted from mammalian cells.** Given the ease of producing recombinant BioSTING and its nanomolar affinity for 2′3′-cGAMP, we anticipated that it may provide an attractive method for measuring 2′3′-cGAMP from cellular extracts. To demonstrate this utility, we monitored FRET responses of BioSTING exposed to methanol extracts from HEK293T cells transfected with increasing concentrations of pCDNA3.1-cGAS. As expected, FRET responses were negligible with low concentrations of transfected pCDNA3.1-cGAS plasmid but saturated at elevated plasmid concentrations (Fig. 2e). Notably, at lower levels of transfected plasmid we observed marginal FRET responses. This is likely a consequence of the simultaneous increase in both cGAS expression and activating DNA, as well as the dilution factor chosen for the experiment. To overcome this limitation, samples containing low levels of cGAMP can be diluted less or subjected to a cGAMP enrichment step using recombinant STING.

In addition, we observed signal saturation with higher levels of transfected plasmid. To overcome signal saturation, samples transfected with the highest amount of pCDNA3.1-cGAS were diluted to obtain readings within the sensor's linear range (Fig. 2f). Multiplying the dilution factor by the diluted sample's cGAMP concentration calculated by interpolation into a standard curve of known cGAMP concentrations (Fig. 1d), thus allows for determination of cGAMP in the undiluted sample. If cell volume assumptions are applied, an average intracellular cGAMP concentration can be calculated, in our case ~30 μM, which was in line with the concentration determined by EIA and within the range of concentrations previously reported by other methods (Supplementary Fig. 2d)[48]. Together, these findings suggest that recombinant BioSTING can be used as a bulk cell extract 2′3′-cGAMP detection method; although, other sensitive methods for end-point 2′3′-cGAMP detection exist[22,48].

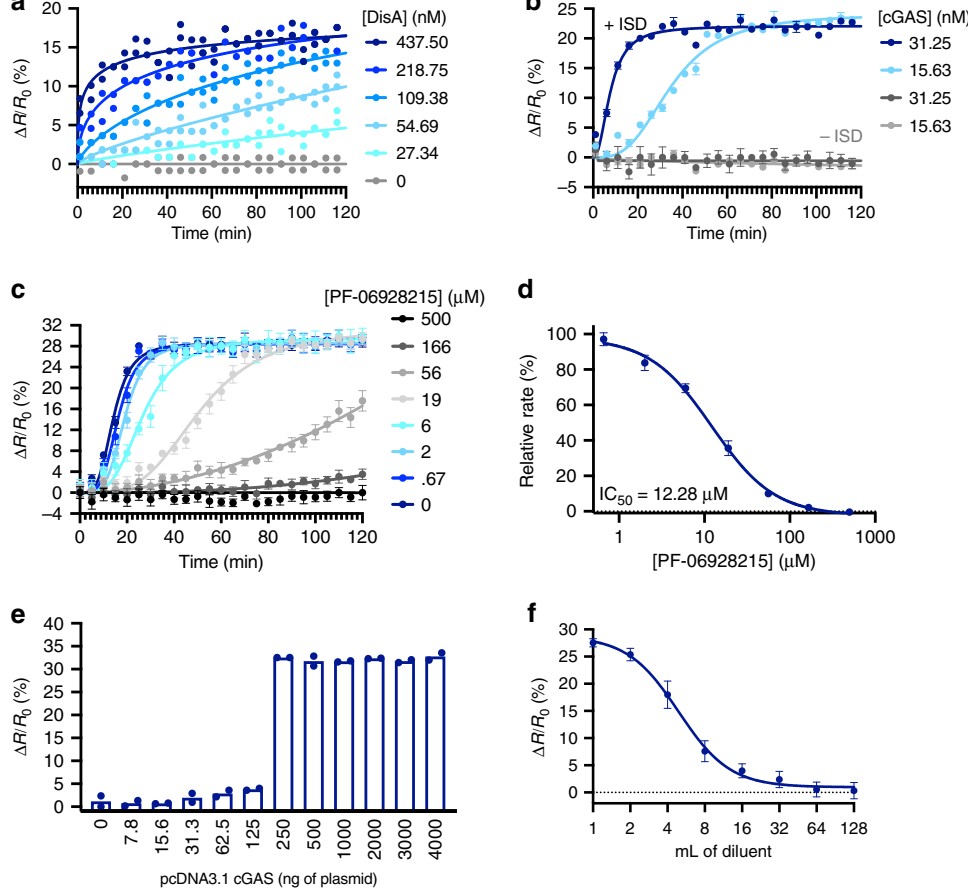

**Fig. 2 Real-time measurement of CDN synthesis and determination of CDN levels from cellular extracts. a** DisA enzyme activity assay time course in the presence of increasing concentrations of the 3'3'-c-di-AMP cyclase DisA using BioSTING. **b** cGAS activity assay in the presence of indicated concentrations of recombinant cGAS with or without Interferon Stimulatory DNA (ISD) using BioSTING. **c** cGAS activity assay measuring 2'3'-cGAMP production in the presence of fixed cGAS and ISD concentrations with increasing concentrations of the cGAS inhibitor PF-06928215 using BioSTING. **d** Reaction linear rates calculated from panel **c** plotted versus PF-06928215 concentration and fit to IC50 curve (Prism 8). **e** BioSTING FRET response in the presence of methanol extracted cGAMP from HEK293T cells transfected with increasing concentrations of pCDNA3.1-cGAS. **f** BioSTING FRET response over two-fold serial dilutions of cGAMP methanol extracts from HEK293T cells transfected with 5 µg of pCDNA3.1-cGAS. In panels **a** and **e**, individual data points of $n = 2$ biological replicates are shown. In panels **b**–**d** and **f**, data are presented as mean ± s.d. of $n = 3$ (**b**, **f**) or $n = 8$ (**c**–**d**) biological replicates.

**BioSTING can detect 2'3'-cGAMP within live mammalian cells**. The ability to directly monitor cyclic dinucleotides in individual living eukaryotic cells is currently one of the biggest limitations in the field. Encouraged by the in vitro characterization of BioSTING, we next sought to extend its application within this context. BioSTING was introduced into a pSLIK doxycycline inducible lentiviral system to generate stable HEK293T cell lines (Supplementary Fig. 3a)[49]. Fixed levels of either pCDNA3.1-cGAS or empty vector were transfected into cells expressing BioSTING and FRET signal was monitored using flow cytometry. In cells expressing cGAS relative to an empty vector control, we observed a greater than 20% BioSTING FRET increase, consistent with our in vitro assays (Fig. 3a–c, Supplementary Fig. 3b, c). We also confirmed production of 2'3'-cGAMP in this experimental system by EIA analysis (Fig. 3b). To ensure that BioSTING FRET responses were directly due to cGAMP recognition, we delivered purified cGAMP using both lipofection and nucleofection. Lipofectamine transfection produced a strong response, which plateaued below the BioSTING saturation level, likely as a consequence of the limited carrying capacity of the transfection reagent (Supplementary Fig. 3d). Nucleofection, on the other hand, was not limited in this manner, and we observed complete BioSTING activation (Fig. 3d).

As intracellular cGAMP concentrations can range from low nanomolar to high micromolar[48], to quantitatively contextualize the observed BioSTING FRET response in cells, we titrated the levels of cGAS expression vector and determined the intracellular concentration of cGAMP by EIA in parallel with BioSTING FRET measurements by flow cytometry. Using assumptions about cell volume, we then plotted FRET changes versus average intracellular cGAMP concentration from which we were able to estimate that 50% of the maximum BioSTING FRET signal corresponds to a concentration of approximately 5–50 nM cGAMP in cells, consistent with the in vitro measured affinity of BioSTING for cGAMP and supporting our expectation that BioSTING is responding to biologically relevant concentrations of cGAMP (Fig. 3e, f).

Reanalysis of our flow cytometry data to detect FRET levels of single cells rather than the entire population confirmed our expectation that, under the conditions tested, there was a unimodal rather than bimodal shift in signal (Supplementary Fig. 3e). Because the entire population is responding with a normal distribution, we interrogated the possibility of using BioSTING as a platform for screening via flow cytometry. To estimate sorting potential, a FRET high gate was drawn above cells transfected with an empty vector and a FRET low gate drawn

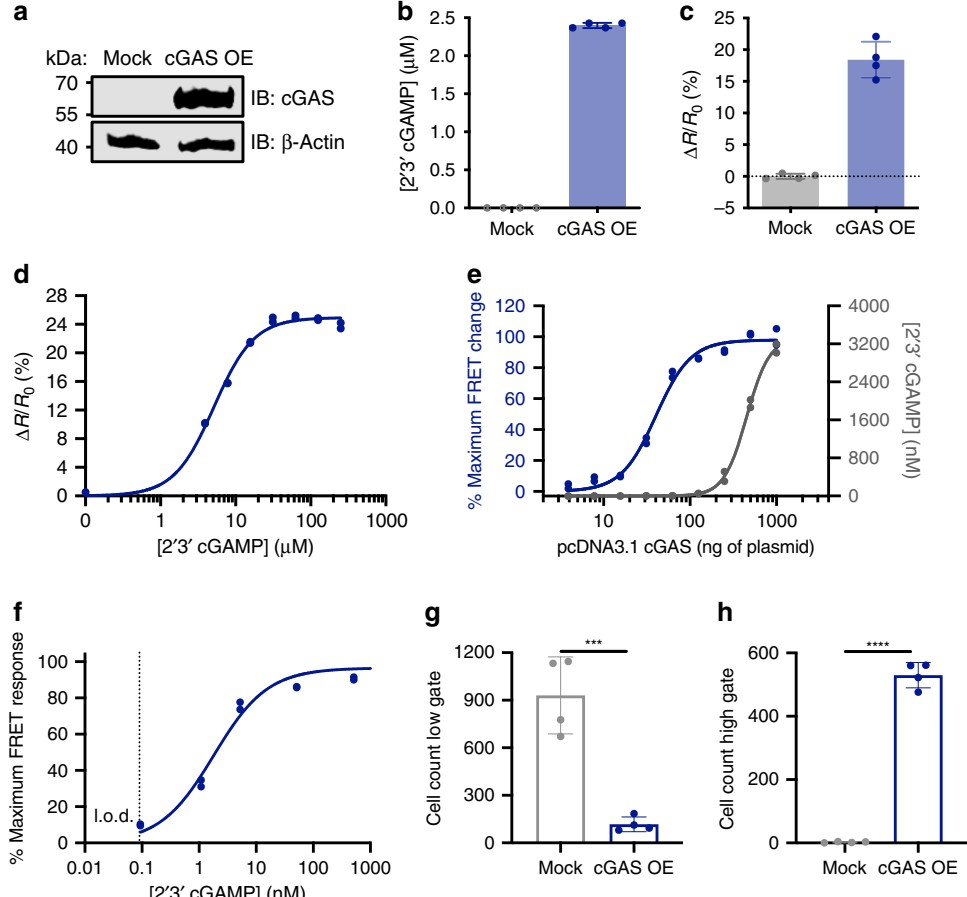

**Fig. 3 BioSTING quantitates cGAMP in single cells in a manner compatible with flow screening.** HEK293T cells stably expressing BioSTING transfected with 1 μg of either empty pCDNA3.1 vector or pCDNA3.1-cGAS vector and analyzed for **a** cGAS expression by western blot (data are representative of two independent experiments), **b** cGAMP production by EIA analysis, and **c** BioSTING FRET response by flow cytometry. **d** HEK293T cells stably expressing BioSTING were electroporated using AMAXA (Lonza) with increasing concentrations of purified 2'3'-cGAMP and analyzed for FRET response by flow cytometry. **e** HEK293T cells stably expressing BioSTING were transfected with increasing concentrations of pCDNA3.1-cGAS and analyzed for FRET response by flow cytometry or for 2'3'-cGAMP production by EIA. **f** BioSTING FRET response from (**e**) graphed as a function of intracellular 2'3'-cGAMP concentration measured by EIA. **g–h** HEK293T cells stably expressing BioSTING transfected with either empty-pCDNA3.1 vector or pCDNA3.1-cGAS vector and analyzed by the alternative gating method in S2d for (**g**) cells in the low gate and (**h**) cells in the high gate. Statistical analyses were performed using two-tailed $t$-tests: *** denotes $P = 0.0006$ (**g**) and **** denotes $P = 0.0000002$ (**h**) (Prism 8). In panels **b–c** and **g–h**, data are presented as mean ± s.d. of $n = 4$ biological replicates. In panels **d–f**, individual data points of $n = 2$ biological replicates are shown.

below cells transfected with a high amount of pCDNA3.1-cGAS (Supplementary Fig. 3f). We obtained ~10-fold enrichment for cells in the FRET low gate (Fig. 3g) and ~100 fold enrichment of cells in the FRET high gate (Fig. 3h). We hypothesize that the decreased selection in the FRET low gate is a consequence of a small percentage of cells evading efficient transfection. Thus, efficiency of selection could likely be improved using an integrated inducible or constitutive cGAS system. Overall, these encouraging results provide support for utilizing BioSTING as a platform for forward genetic screens to identify genes involved in regulating cGAS and cGAMP in living cells.

In order to assess the impacts of BioSTING expression levels on intracellular FRET responses, we again reanalyzed our flow cytometry results to compare cells expressing high and low levels of BioSTING (Supplementary Fig. 4a). These two populations had similar FRET response dynamics but with slightly different response magnitudes (Supplementary Fig. 4b–d). Thus, although there is flexibility in the expression level of cells, BioSTING expression levels should be tightly controlled when comparing FRET responses between samples.

**Development and use of a cGAMP-blind BioSTING.** A powerful control for an intramolecular FRET biosensor is a ligand blind version which differentiates changes due to bonafide binding from other effects (i.e., protein–protein interactions, fluorophore quenching, etc.). A literature search highlighted the mutations Y240S and T263A in human STING (Y239S and T262A in murine STING) as ideal candidates which would diminish CDN binding without disrupting protein stability (Fig. 4a)[50]. As expected, Y239S T262A BioSTING stably expressed and neither appreciably bound radioactive cGAMP nor produced a FRET change in response to cGAMP (Fig. 4b, c). To determine if Y239S T262A BioSTING could act as a cGAMP control in cells, we titrated pCDNA3.1-cGAS in cells expressing WT or mutant biosensor. While WT BioSTING produced a robust FRET response, Y239S T262A BioSTING only generated a minor FRET change at exceptionally high levels of cGAS, thus supporting the use of Y239S T262A BioSTING as a control biosensor for 2'3'-cGAMP (Fig. 4d).

Bacterial 3'3'-CDNs have been shown to be released during infection and to activate STING in the cytosol. To determine if

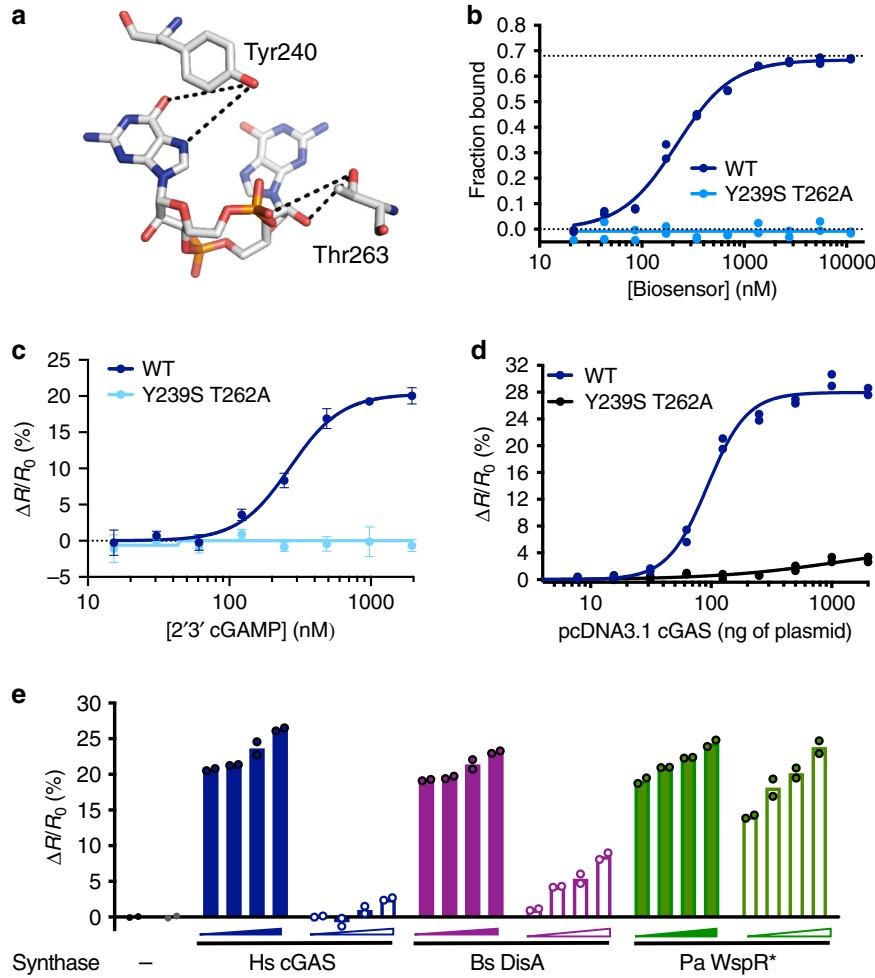

**Fig. 4 BioSTING variants exhibit distinct specificity for metazoan and bacterial CDNs. a** Interactions made by human STING residues Y240 (mouse Y239) and T263 (mouse T262) hypothesized to stabilize CDN binding in PDB 4F5D visualized in PyMol. **b** DRaCALA binding analysis of WT and Y239S T262A BioSTING using [$^{32}$P] labeled 2'3'-cGAMP. **c** Recombinant WT and Y239S T262A BioSTING FRET response in the presence of increasing concentrations of 2'3'-cGAMP. Data are presented as mean ± s.d. of $n = 4$ biological replicates. **d** HEK293T cells stably expressing WT or Y239S T262A BioSTING were transfected with increasing concentrations of pCDNA3.1-cGAS and analyzed for FRET response by flow cytometry. **e** HEK293T cells stably expressing WT (solid bars) or Y239S T262A (open bars) BioSTING were transfected with increasing concentrations (125, 250, 500, or 1000 ng) of expression vectors for cGAS, DisA, WspR*, or empty vector and analyzed for FRET response by flow cytometry. Data in panels **b**, **d** and **e** are presented as individual data points of $n = 2$ biological replicates are shown.

WT BioSTING and Y239S T262A BioSTING can detect bacterial CDNs in cells, we titrated increasing concentrations of expression vectors for *B. subtilis* DisA which synthesizes c-di-AMP and a constitutively active isoform of *P. aeruginosa* WspR (D70E), denoted WspR*, which synthesizes c-di-GMP (Fig. 4e)[40]. All nucleotides resulted in a BioSTING FRET increase. However, while Y239S T262A BioSTING produced a minor response to the highest concentrations of cGAS, moderate responses were observed for DisA, and unexpectedly large responses were observed for WspR*. Levels of c-di-AMP are not expected to reach high levels in physiologically relevant contexts, therefore Y239S T262A BioSTING is likely to serve as an adequate control for c-di-AMP. In contrast, the lowest levels of WspR* tested induced a large response in Y239S T262A BioSTING demonstrating that it will not be an adequate control for c-di-GMP. Thus, despite its applicability as a control for other CDNs, the Y239S T262A BioSTING variant may be selectively used as a c-di-GMP biosensor within cells.

We hypothesized that further design may be used to generate other versions of BioSTING which exhibit selectivity for CDNs based upon their unique chemical properties, including

phosphodiester linkage and/or base content. Previous studies identified naturally occurring mutations in human STING that abolish responsiveness to 3'3'-CDNs while retaining 2'3'-cGAMP sensing[8]. In an effort to engineer a universally blind control biosensor we subsequently made the R231A/H mutations previously reported to diminish IFN-β activation in Y239S T262A BioSTING. We also introduced these mutations to WT BioSTING in an effort to make a sensor capable of uncoupling bacterial versus eukaryotic CDNs. When cyclic dinucleotide cyclases are expressed in this mutant array, we unexpectedly found that these mutants alone only diminished the response to cGAMP but the triple mutant Y239S, T262A, and R231A diminished but did not completely abrogate the FRET response to c-di-GMP (Supplementary Fig. 5). More work will be required to determine whether this is related to 3'3'-c-di-GMP binding before this sensor could be employed as a control for c-di-GMP secretion. Therefore, we were unsuccessful in our attempt to uncouple sensing of 3'3'-CDNs from 2'3'-cGAMP. In future iterations of BioSTING, we hope to use unbiased approaches to make BioSTING variants with altered nucleotide specificities.

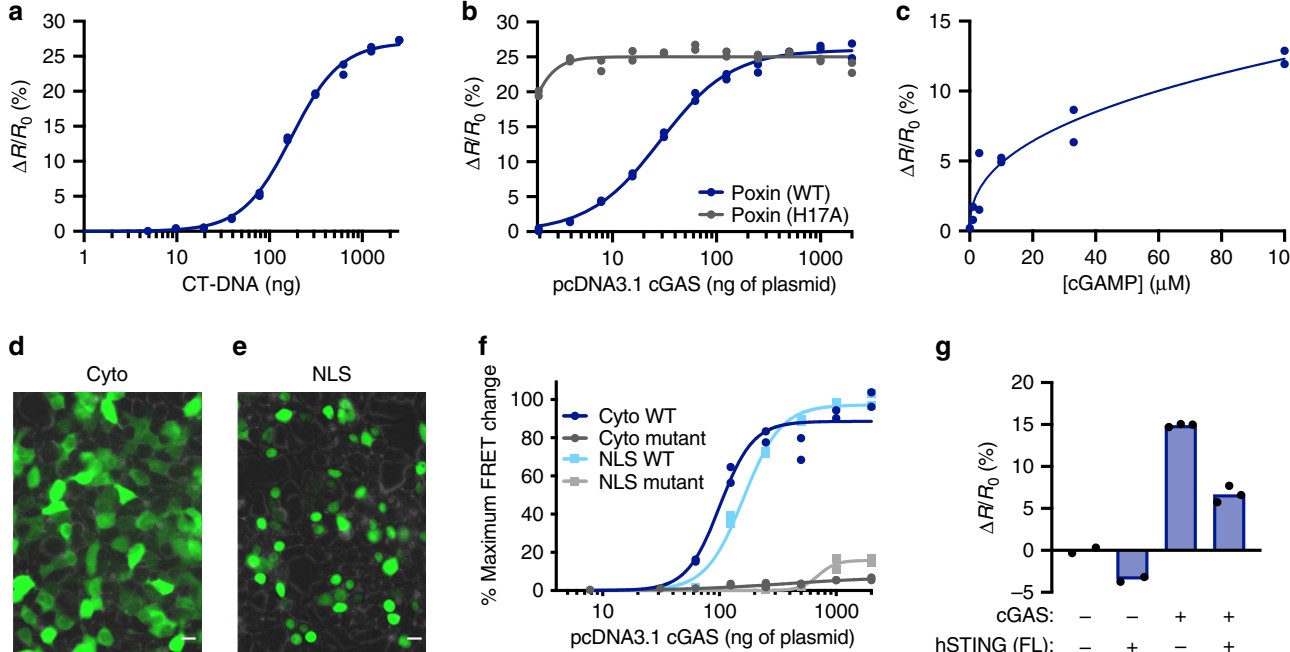

**Fig. 5 BioSTING exhibits broad utility for monitoring diverse aspects of cGAMP signaling. a** HEK293T cells stably expressing BioSTING were transfected with 10 ng of pCDNA3.1-cGAS for 16-20 h then transfected again with an increasing concentration of activating CT-DNA for 4 h and analyzed by flow cytometry. **b** HEK293T cells stably expressing BioSTING were transfected with 3 μg of WT or H17A Poxin and increasing concentrations of pCDNA3.1-cGAS and analyzed for FRET response by flow cytometry. **c** Increasing concentrations of 2'3'-cGAMP were added to the media of HEK293T cells stably expressing BioSTING for 6 h and analyzed for FRET response by flow cytometry. **d-e** BioSTING expression was induced in HEK293T cells transduced with pSLIK-BioSTING or pSLIK-NLS-BioSTING for 24 h by the addition of doxycycline and biosensor expression was analyzed using a BZ-X710 microscope (Keyence) using brightfield and a GFP filter cube (EX 470/40, DM 495, and BA 525/50) with a 40X objective. Scale bars, 15 μm. Data are representative of two independent experiments. **f** HEK293T cells stably expressing untagged or NLS-tagged WT or Y239S T262A BioSTING were transfected with increasing concentrations of pCDNA3.1-cGAS and analyzed for FRET response by flow cytometry. **g** HEK293T cells stably expressing untagged WT BioSTING were transfected with 100 ng of cGAS expression plasmid or empty vector alone with either 100 ng of empty vector or vector expressing full-length (FL) human STING and analyzed by flow cytometry. In all panels, individual data points of $n = 2$ (**a-c**, **f-g**) or $n = 3$ (**g**) biological replicates are shown.

**BioSTING exhibits broad utility for monitoring 2'3'-cGAMP dynamics in live cells.** Based on our encouraging results highlighting BioSTING's ability to detect cGAMP in cells, we next sought to demonstrate BioSTING's utility to monitor modulation of cGAS activity. Previous experiments titrated pCDNA3.1-cGAS alone, resulting in simultaneous increases in both the CDN cyclase and stimulatory ligand, leading to titration curves with positive Hill coefficients. To directly measure cGAS activation we co-transfected cells with a fixed, low concentration of pCDNA3.1-cGAS and increasing amounts of calf thymus DNA (CT-DNA) ligand. As expected, at low CT-DNA levels we observed no detectable FRET increase, but as CT-DNA content was increased, we observed an elevated FRET signal that began to saturate at the highest concentration of stimulatory ligand tested (Fig. 5a). These results suggest that BioSTING is a useful tool to investigate the kinetics of cGAS activation in live cells.

The cGAS-STING pathway is immensely important for controlling viral infection and many viruses have developed methods to inhibit this pathway. Recently, vaccinia virus was reported to antagonize cGAS-STING signaling through expression of Poxin, a 2'3'-cGAMP-selective phosphodiesterase[51]. To monitor cGAMP hydrolysis by Poxin in cells, we expressed a constant level of wild-type (WT) or catalytically dead (H17A) Poxin over a titration of cGAS plasmid. Expression of Poxin in 293T cells was confirmed by [$^{32}$P] CDN hydrolysis assays (Supplementary Fig. 6a, b). In cells transfected with low levels of cGAS plasmid we observed high FRET from the H17A mutant and a greatly decreased FRET response in cells expressing the WT

Poxin. As expected, expression of high cGAS levels overcame the capacity of Poxin to antagonize cGAMP levels (Fig. 5b). Co-expression of Poxin with DisA or WspR* had no effect on 3'3'-CDN-mediated FRET responses as compared to empty vector controls, consistent with the role of Poxin as a 2'3'-cGAMP specific hydrolase (Supplementary Fig. 6c). These results demonstrate the utility of BioSTING to characterize modulators of cGAMP concentrations within living cells.

While cGAS production of cGAMP and subsequent STING activation both occur in the cytosol, recent findings have established that CDNs can be transmitted between cells through export and import mechanisms, as well as through gap junctions[20,21,48,52,53]. Such nucleotide transfer is reported to facilitate antitumor properties and cGAMP is being explored therapeutically both alone and in combination with PD-1 blockade[48,52–56]. The therapeutic utility of cGAMP in this context requires nucleotide import to the cytosol to promote STING inflammatory responses[20,48,52–54]. To investigate the utility of BioSTING for monitoring cellular import of cGAMP, we added increasing concentrations of cGAMP to the extracellular medium. After 6 h, cellular FRET signals in response to altered concentrations of cGAMP exhibited saturation like responses below the maximum signal associated with the sensor (Fig. 5c). The observed saturation of the response may be due to saturation of the importer operative in these cells or a consequence of the establishment of import-export equilibrium. While a thorough account of the transport mechanisms is yet to be documented, these results demonstrate that BioSTING can

detect cGAMP uptake from the extracellular space and may provide a valuable tool to identify and characterize the mechanism by which cGAMP is transferred among cells.

Despite being a well-studied system, conflicting reports regarding cGAS and cGAMP localization remain[57–61]. As a genetically encodable protein, we hypothesized that BioSTING could be localized to distinct cellular compartments through the introduction of specific signal sequences. As such, we introduced a nuclear localization signal (NLS) to BioSTING, which resulted in successful localization to the nucleus (Fig. 5d, e, Supplementary Fig. 6d, e). Using a cGAS titration, we were able to detect cGAMP in both the nucleus and the cytoplasm (Fig. 5f). Consistently, the Y239S T262A BioSTING control sensor had very little change in FRET response providing additional evidence that we are detecting bonafide cGAMP in the nuclear compartment. Although these experiments only indicate that cGAMP can penetrate the nucleus, we anticipate that utilization of BioSTING in time-course and microscopy experiments will further elucidate the localization of cGAMP production under various activating conditions and may be used to reveal mediators of nucleotide transit between cellular compartments.

Finally, BioSTING, containing the dimerization domain of full-length STING, has the potential to heterodimerize with endogenous STING. To determine the consequences of heterodimerization, full-length, human STING was expressed alone or in combination with cGAS in HEK293T cells stably expressing BioSTING. Expression of human STING in this context decreased the response of BioSTING in the presence and absence of cGAS (Fig. 5g). Observation of a FRET decrease upon expression of STING suggests that heterodimerization leads to an altered conformation and that cells must be deficient for STING in order to attain interpretable FRET responses. Taken together, these data demonstrate that our first generation FRET biosensor, BioSTING, is highly versatile for both in vitro and cellular studies, but its application is currently limited to a STING-deficient setting.

## Discussion

Here, we report the development of BioSTING, a FRET-based intramolecular biosensor engineered to monitor CDNs in vitro and in cells. BioSTING maintains the native CDN binding properties of the parent protein upon which it was designed and as such exhibits CDN reporting capacity in physiological concentration ranges. Through a variety of in vitro and cellular studies we establish BioSTING as a robust sensor of a variety of CDNs, providing real-time detection of nucleotide levels with temporal and spatial resolution.

BioSTING's ease of recombinant production, native binding properties, and simple kinetic readout make it a promising tool for investigating cyclic dinucleotides in vitro. We show that BioSTING can detect extracted nucleotides from cellular samples, as well as enzymatic production of cyclic dinucleotides with recombinant protein. Although not investigated in this work, BioSTING can also likely be used to monitor phosphodiesterase activity for bacterial 3'3'-cyclic dinucleotides in real time. However, given the unexpectedly low dissociation rate, such application for 2'3-cGAMP hydrolysis may be limited. In addition to general enzymatic characterization of CDN synthesis, we also utilized BioSTING to characterize PF-06928215 inhibition of cGAS, demonstrating its robust utility for characterization of small molecule modulators of CDN synthases. A key limitation to be considered relates to the spectral properties of compounds under investigation. For instance, while several antimalarial compounds have been reported to inhibit cGAS activity, the intrinsic fluorescence of these compounds spectrally overlap with BioSTING and interfere with its application in this context.

Despite impressive in vitro utility, the primary motivation for developing BioSTING was to create a tool capable of detecting 2'3'-cGAMP in live single cells. While the dynamic range of BioSTING is indeed narrower, our findings support that BioST-ING binds and responds to cGAMP at physiologically relevant concentrations and with sensitivity comparable to commercially available EIA based approaches. In addition, we successfully measured FRET changes by flow cytometry, providing the first single cell measurements of CDN levels. In addition to single cell measurements, a key aspect of FRET based biosensors is the capacity to provide subcellular information about signaling dynamics. As a genetically encodable protein, localization tags can be added to target a sensor to specific cellular compartments. There is currently a debate about the localization of cGAS and therefore production of cGAMP in cells. By introducing an NLS to BioSTING we demonstrated the ability to restrict the sensor to the nucleus and monitor cGAMP within this compartment. Application of BioSTING in combination with rapid imaging microscopy is likely to provide important insight into when and where cGAMP is produced and if this differs depending upon the infectious insult, within distinct cell types, or in response to cellular damage versus pathogen encounter.

One particularly exciting application of BioSTING is for high throughput small molecule and forward genetic screening. In high throughput small molecule screening applications, BioST-ING provides the ability to measure cGAMP production in live cells, which can simultaneously account for compound toxicity and cell permeability together with target engagement. In fact, Pfizer identified PF-06928215 as an inhibitor of cGAS through in vitro enzyme screening methods but the compound failed in development as a pharmaceutical due to its inability to access the cytosol. In addition, we anticipate that BioSTING will have utility in genetic screens to study modulation of cGAMP levels in cells. Because FRET measurements can be conducted using flow cytometry, BioSTING affords the ability to conduct these studies in batch culture. Combining BioSTING flow-based sorting with disruption or overexpression libraries from mammalian and microbial pathogens will afford genome wide interrogation of CDN signaling pathways, including molecular insight into pathogen associated antagonists, as well as cell intrinsic regulators of the pathway, namely cGAS activators and inhibitors and mediators of cGAMP hydrolysis, transport, and localization.

Though developed to investigate 2'3'-cGAMP cellular biology, BioSTING is also capable of investigating similar aspects of bacterial 3'3'-cyclic dinucleotides. We demonstrated through expression of DisA and WspR* that BioSTING can detect bacterial 3'3'-CDNs in the mammalian cytosol. Thus, the potential for BioSTING to be used as a tool to investigate the timing and magnitude of bacterial CDN release in biologically relevant contexts such as during *Listeria monocytogenes, Mycobacterium tuberculosis,* and *Chlamydia trachomatis* infection, among others, is evident. In our attempts to identify CDN blind variants of BioSTING, we inadvertently identified Y239S T262A BioSTING as a c-di-GMP specific reporter. These findings suggest that with further engineering of BioSTING it may be possible to identify mutants sensitive to specific CDNs. Such sensors would be useful to dissect mixed CDN interactions such as during *M. tuberculosis* infection, in which both cGAS produced 2'3'-cGAMP and bacterial derived c-di-AMP have been implicated in STING activation[62–64]. In addition, a wide array of bacterial cyclic dinucleotides of dipurine, dipyrimidine, and mixed purine pyrimidine content were recently reported; however, only cyclic dipurine containing nucleotides were shown to robustly bind to and activate STING[14,65]. It is feasible that BioSTING can be engineered through a mix of semi-randomized and rationally designed mutations to detect these more recently discovered

signaling nucleotides. Finally, mutation of Tyrosine 167, which stabilizes CDN binding though a pi-stacking interaction, to an Alanine or Valine will likely generate an additional BioSTING control variant that is universally blind to all CDNs. Although outside the scope of this work, the intracellular concentration of bacterial CDNs have been reported to be similar to the Kd of BioSTING[13,32,64], thus it is feasible that expression of BioSTING in bacterial cells will afford interrogation of CDN dynamics within these organisms as well.

BioSTING is a blue, orange-based rather than far-red or luminescence-based biosensor, which limits its use to cell culture due to the limited ability for blue-orange light to penetrate tissues. Although there are many important findings to be made in cell culture, as a clinically relevant molecule, there is also an immense interest in studying 2'3'-cGAMP in vivo. We believe that it will be possible to develop a far-red FRET or luminescent version of BioSTING by replacing mKO2-mTFP fluorophores with a small circularly rotated library of a red-shifted FRET pair, BRET pair, or split luciferase and screening for increased signal upon cGAMP binding[66–70]. Production of either a far-red or luminescent derivative integrated into the murine genome under either constitutive or cell specific promoters will allow for the detection of cGAMP in vivo and create a powerful model to study cGAMP activity in viral infection and autoimmune disease models.

As a STING-based biosensor, BioSTING is sensitive to heterodimerization with native STING. Thus, the sensitivity afforded by STING also leads to the limitation that BioSTING must be used in cells which either do not express or are genetically modified to lack STING. Although many important studies are tractable in this system, some investigations such as those coupling cGAMP measurements with STING-Interferon pathway activation or regulation in the same sample currently are not. To allow investigation of these exciting research areas we anticipate creating a version of BioSTING incapable of heterodimerization with WT STING through engineering of the dimerization interface. This updated version will increase BioSTING's versatility while retaining the impressive sensitivity and specificity of STING.

Overall BioSTING is a powerful tool that makes many fundamental and clinically important investigations of cyclic dinucleotide biology more tractable and in some instances even feasible. In addition to the immediate application of current BioSTING versions, we believe there is immense promise in using BioSTING as a foundation to develop a wide array of biosensors with unique CDN binding or in vivo imaging capacities. In total, BioSTING represents a versatile tool to significantly advance the current limits of knowledge related to the ever expanding and clinically relevant field of cyclic dinucleotide signaling.

## Methods

**BioSTING cloning**. Primers for BioSTING cloning are listed in Supplementary Table 2 and plasmids and strains are listed in Supplementary Table 3. Prototype and GGSGG linker versions of BioSTING were generated by amplifying STING CTD with Kapa HiFi polymerase (Kapa Biosystems) using a combination of primers 1, 2, 3, and 4 from pET28b-mSTING CTD. The resulting products were ligated into pET15b-mKO2-12AA-mTFP using Spe1/Kpn1 fast digest restriction endonuclease cloning (Thermo Fisher) and transformed into XL1-Blue chemically competent E. coli. Site directed mutagenesis was carried out by amplifying the generated pET15b-BioSTING sensor using primers 5, 6 or 7, 8 or 9, 10, or 11,12 using Kapa HiFi polymerase. PCR purified product was DpnI digested (NEB) and transformed into XL1-Blue chemically competent E. coli. To generate pSLIK-BioSTING, pET15b-BioSTING was amplified using primers 13 and 14 for cytoplasmic expression and primers 13 and 15 to add a nuclear localization signal (NLS) from c-MYC at the C-terminus. These products were then ligated into the BsiW1 (Thermo Fisher) site of pSLIK using InFusion (Takara) then transformed into Stbl3-OneShot competent cells (Thermo Fisher).

**Protein expression and purification**. Recombinant 6×-His tagged SUMO-mcGAS, B. subtilis (B.s.) DisA, mSTING-CTD, and mRECON were expressed and purified as summarized below[20]. Briefly, plasmids for mcGAS, DisA, mSTING-CTD, and mRECON expression were transformed into Rosetta (DE3)pLysS chemically competent cells. Overnight cultures of the resulting transformed bacteria were inoculated into 1.5 L of LB broth at a 1:100 dilution. Bacterial cultures were grown to OD_600 0.4–0.6 at 37 °C after which protein expression was induced by the addition of 0.5 mM isopropyl β-D-1-thiogalactopyranoside (IPTG) for 20 h at 16 °C. Bacteria were harvested by centrifugation, and the cell pellets were resuspended in Buffer A [50 mM Tris-Cl pH = 8.0, 300 mM NaCl, 20 mM Imidazole, 5 mM β-Mercaptoethanol (BME), and 1 mM phenylmethylsulfonyl fluoride (PMSF)]. Cells were lysed by sonication and clarified lysate was bound to HisPur NiNTA Resin (Thermo Scientific). The resin was washed with 100–200 column volumes of buffer A and bound proteins were eluted in Buffer B [50 mM Tris-Cl pH = 7.4, 300 mM NaCl, 300 mM Imidazole, 5 mM β-Mercaptoethanol (BME), and 1 mM phenylmethylsulfonyl fluoride (PMSF)]. Following NiNTA chromatography, His6-SUMO-mcGAS was exchanged into Buffer C [20 mM Tris-Cl pH 7.4, 250 mM NaCl, 1 mM dithiothreitol (DTT)] and further purified by Heparin Sepharose chromatography. Bound cGAS was eluted over 250 mM to 1000 mM NaCl gradient. The resulting purified proteins were analyzed by SDS-PAGE, exchanged into storage buffer [40 mM Tris pH 7.5, 100 mM NaCl, 20 mM MgCl_2] using PD-10 desalting columns (GE Healthcare), snap frozen, and stored at −80 °C until use.

For BioSTING expression, plasmids encoding BioSTING variants were transformed into Rosetta (DE3)pLysS chemically competent cells. Overnight cultures of the resulting transformed bacteria were inoculated into 1 L of LB broth and grown as above. At an OD_600 of 0.5–0.7, protein expression was induced by the addition of 0.2 mM isopropyl β-D-1-thiogalactopyranoside (IPTG) for 6 h at 18 °C. Bacteria were harvested by centrifugation, and the cell pellets were resuspended in Buffer D [50 mM Tris-Cl pH 7.5, 100 mM NaCl, 20 mM Imidazole, 5 mM β-Mercaptoethanol (BME), and 1 mM phenylmethylsulfonyl fluoride (PMSF)]. Cells were lysed by sonication and clarified lysate was bound to HisPur NiNTA Resin (Thermo Scientific, Waltham, MA). The resin was washed with 100–200 column volumes of Buffer D and bound proteins were eluted in Buffer E [50 mM Tris-Cl pH = 7.5, 100 mM NaCl, 300 mM Imidazole, 5 mM β-Mercaptoethanol (BME), and 1 mM phenylmethylsulfonyl fluoride (PMSF)]. The resulting proteins were concentrated and further purified by gel filtration on a Superdex 200 column (GE Healthcare) using storage buffer [40 mM Tris pH 7.5, 100 mM NaCl, 30 mM MgCl_2, supplemented with 0.5 mM TCEP]. Protein samples were tested for purity by SDS-PAGE followed by Coomassie Brilliant Blue staining. Fractions with high purity were pooled, concentrated, flash frozen, and stored at −80 °C until use in biochemical assays.

**Synthesis of [³²P] 2'3'-cyclic GMP-AMP and [³²P] 3'3'-cyclic di-AMP**. [³²P] Radiolabeled cyclic dinucleotides (CDNs) were synthesized enzymatically using α-[³²P] ATP (Perkin-Elmer) and recombinant SUMO-mcGAS (2'3'-cGAMP) or B.s. DisA (3'3'-c-di-AMP) and affinity purified using mSTING-CTD and mRECON, as follows:

[³²P] cGAMP was synthesized enzymatically by incubating 0.33 μM α-[³²P] ATP (Perkin-Elmer) with 250 μM unlabeled GTP, 1 μg of Interferon Stimulatory DNA 100mer, and 1 μM of recombinant His-tagged cGAS in binding buffer [40 mM Tris pH 7.5, 100 mM NaCl, 20 mM MgCl_2] at 37 °C overnight. Subsequently, recombinant cGAS was removed from the reaction mixture by incubation with HisPur Ni-NTA resin (Thermo Scientific) for 30 min. The sample was transferred to a minispin column (Thermo Scientific) to elute the crude [³²P] cGAMP sample. The resulting [³²P] cGAMP was purified further using recombinant mSTING-CTD. 100 μM mSTING-CTD was bound to HisPur Ni-NTA resin for 30 min on ice. The resin was washed two times to remove unbound mSTING-CTD. The resulting resin was incubated with the remaining crude cGAMP synthesis reaction mixture for 30 min on ice. Following removal of the supernatant, the Ni-NTA resin was washed five times with ice cold binding buffer. The resin was then incubated with 100 μL of binding buffer for 10 min at 95 °C and transferred to a minispin column to elute [³²P] cGAMP.

[³²P] c-di-AMP was synthesized as follows: briefly, 1 μM α-[³²P] ATP (Perkin-Elmer) was incubated with 1 μM of recombinant DisA in binding buffer at 37 °C overnight. The reaction mixture was boiled for 5 min at 95 °C and DisA was removed by incubation with HisPur Ni-NTA resin. The sample was transferred to a minispin column to elute the crude [³²P] c-di-AMP sample. The resulting [³²P] c-di-AMP was further purified using recombinant His-tagged mRECON. 100 μM His-tagged mRECON was bound to HisPur Ni-NTA resin for 30 min on ice. The resin was washed two times to remove unbound RECON. The resulting resin was incubated with the remaining crude [³²P] c-di-AMP sample for 30 min on ice. Following removal of the supernatant, the Ni-NTA resin was washed five times with ice cold binding buffer and then incubated with 100 μL of binding buffer for 5 min at 95 °C. The slurry was then transferred to a minispin column to elute [³²P] c-di-AMP.

Affinity purified CDNs were analyzed by Thin Layer Chromatography (TLC) on Polygram CEL300 PEI TLC plates (Machery-Nagel) in buffer containing 1:1.5 (vol/vol) saturated $(NH_4)_2SO_4$ and 1.5 M $NaH_2PO_4$ pH 3.6. [³²P] radiolabeled CDNs were visualized by exposure onto PhosphorImager screens, which were developed using a Typhoon FLA 9000 biomolecular imager (GE Healthcare) and determined to be ~99% pure.

**Nucleotide binding assays.** [32P] Radiolabeled cyclic dinucleotide binding assays were performed using DRaCALA[46]. Binding assays were performed in binding buffer [40 mM Tris pH 7.5, 100 mM NaCl, 20 mM MgCl$_2$] at room temperature. To determine binding affinities, two-fold serial dilutions of proteins were incubated with ~1 nM of [32P] radiolabeled CDNs for at least 10 min. To determine binding specificities, proteins were pre-incubated with 500 µM excess, unlabeled nucleotides for 10 min, followed by incubation with ~1 nM of [32P] radiolabeled CDNs for at least 10 min. Samples were then blotted onto nitrocellulose membranes and allowed to air dry. [32P] radioactivity was visualized by exposure onto Phosphor-Imager screens, which were developed using a Typhoon FLA 9000 biomolecular imager (GE Healthcare). Non-radioactive 2'3'-cGAMP (Invivogen), 3'3'-cGAMP (Invivogen), 3'3'-c-di-AMP (Invivogen, San Diego, CA), and 3'3'-c-di-GMP (BIOLOG Life Science Institute, Bremen, Germany) were purchased and diluted in endotoxin free water.

In vitro FRET assays. 5–10 µM purified BioSTING proteins were incubated with increasing concentrations of cyclic dinucleotides within a black flat bottom opaque 96-well plate (Greiner Bio-One) in activity buffer [40 mM Tris pH 7.5, 100 mM NaCl, 20 mM MgCl$_2$]. mTFP and FRET fluorescence was monitored using a fluorimeter (BioTek Synergy H1 Hybrid Reader, BioTek Instruments) with the following parameters (unless otherwise stated): 458 nm excitation, 490 nm emission for mTFP, and 600 nm emission for mKO2. Assay parameters were calculated using Prism Software. Kd and EC50 were determined using nonlinear fit and all other parameters were calculated using linear fit, according to literature precedent [26].

**Enzyme activity assays.** For DisA enzyme activity assays, 5–10 µM purified BioSTING was incubated with increasing concentrations of recombinant *B.s.* DisA within a black 96-well plate in activity buffer [40 mM Tris pH 7.5, 100 mM NaCl, 20 mM MgCl$_2$]. Enzyme assays were initiated by the addition of 1 mM ATP, and the enzyme reactions were allowed to proceed for 2 h at 37 °C.

For cGAS enzyme activity assays, 5–10 µM purified BioSTING was incubated with increasing concentrations of recombinant SUMO-mcGAS within a black 96-well plate in activity buffer [40 mM Tris pH 7.5, 100 mM NaCl, 20 mM MgCl$_2$]. Enzyme assays were initiated by the addition of 1 mM ATP, 1 mM GTP and 1 µg ISD (Fig. 2b, Supplementary Fig. 2a, b) or by the addition of 250 µM ATP, 250 µM GTP, and 50 ng ISD (Fig. 2c, d, Supplementary Fig. 2c). Enzyme assays were allowed to proceed for 2 h at 37 °C. cGAS inhibitor (PF-06928215) was purchased from Sigma-Aldrich and diluted in sterile DMSO.

For all assays, FRET activity was monitored as above.

**2'3'-cGAMP extraction.** HEK293T cells were plated at a density of 750,000 cells per well of a 6-well cell culture plate. The next day cells were transfected with the indicated amounts of pcDNA3.1-hcGAS vector using PEI transfection reagent (Polysciences). Twenty four hours later the cells were harvested by centrifugation and washed once with ice-cold PBS. Cell pellets were resuspended in ice-cold 80% Optima, HPLC grade methanol (Fisher Scientific) and incubated on ice for 20 min. Cells were further lysed by sonication. Following centrifugation, cellular extracts were completely dried under vacuum and stored at −20 °C until use. For FRET assays, extracts were resuspended in activity buffer [40 mM Tris pH 7.5, 100 mM NaCl, 20 mM MgCl$_2$] containing 5–10 µM purified BioSTING and transferred to a black 96-well plate. FRET activity was monitored using a fluorimeter as described above. Quantification requires a standard curve of known cGAMP concentrations in the sample buffer done at the time of analysis.

**Cell lines.** Human Embryonic Kidney (HEK) 293 T cells were grown in Dulbecco's Modified Eagle Medium (DMEM) (Gibco) supplemented with 10% (v/v) heat-inactivated FBS (HyClone), 1 mM sodium pyruvate, 2 mM L-Glutamine (Thermo Fisher), 100 U mL$^{-1}$ penicillin, 100 µg mL$^{-1}$ streptomycin and maintained at 37 °C in 5% CO2 in a humidified incubator.

**Lentivirus production and transduction.** VSV-G pseudotyped, self-inactivating lentivirus was prepared by transfecting a semi-confluent 10 cm dish of HEK293T cells with 4 µg of psPAX2, 2 µg of pCMV-VSV-G, together with 4 µg of pSLIK lentiviral vector using Poly(ethyleneimine) (PEI). Growth medium was replaced 24 h after transfection and cell culture supernatants were collected at 48 and 72 h after transfection and filtered through a 0.45 µm filter.

For lentiviral transduction, HEK293T cells were seeded at a density of 2 to 4 million cells per 10 cm dish. The following day, cells were transduced with 5 mL of filtered lentiviral supernatant. 24 h later the cell culture medium was removed and replaced with standard cell culture medium supplemented with 2 µg per mL puromycin (Gibco). For all subsequent experiments, lentivirus-transduced cells were passaged and maintained in selection medium containing puromycin.

**Intracellular FRET assays.** For experiments using CDN cyclases, HEK293T cells stably expressing the indicated BioSTING constructs under a doxycycline-inducible promoter were plated at a density of 750,000 cells per well of a 6-well cell culture plate. The next day, the cells were transfected with the indicated amounts of cyclase-encoding plasmids using PEI transfection reagent. One hour later, biosensor expression was induced by the addition of Doxycycline Hydrochloride (1 µg

mL$^{-1}$) (Sigma-Aldrich). Twenty four hours later the cells were harvested by centrifugation and resuspended in ice-cold PBS. Biosensor activity was determined by FACS analysis. To contextualize results, a negative control in which no cGAMP is present and a positive control where cGAMP is abundant should be run in each assay to quantify the lower and upper bounds of FRET activation.

For electroporation experiments, HEK293T cells stably expressing wild-type BioSTING were plated at a density of five million cells per 10 cm dish in cell culture medium supplemented with Doxycycline (1 µg mL$^{-1}$) to induce biosensor expression. The next day cells were harvested by trypsinization and electroporated with the indicated concentrations of 2'3'-cGAMP using SF Cell Line 4D-Nucleofector X Kit according to the manufacturer's protocols (Lonza). Following electroporation, the cells were resuspended in ice-cold PBS and analyzed by FACS analysis.

For extracellular 2'3'-cGAMP stimulations, HEK293T cells stably expressing wild-type BioSTING were plated at a density 750,000 cells per well of a 6-well cell culture plate in cell culture medium supplemented with Doxycycline (1 µg mL$^{-1}$) to induce biosensor expression. The next day, the indicated concentrations of 2'3'-cGAMP were added to the culture medium. 6 h later the cells were harvested by centrifugation and resuspended in ice-cold PBS. Biosensor activation was determined by FACS analysis.

**Flow cytometry.** To prepare cells for flow cytometry, cell culture media was aspirated, and the cells were harvested in ice-cold PBS. The resuspended cells were then analyzed using a LSR II flow cytometer (BD) with the following voltages: FSC-A/H/W-350, SSC-A/H/W-240, BV510(mTFP)-360, PE(mKO2)-380, and BV570 (FRET)-375 volts. Data was then analyzed using FlowJo software (Tree Star).

**2'3'-cGAMP enzyme immunoassay (EIA).** HEK293T cells were plated at a density of 750,000 cells per well of a 6-well cell culture plate. The next day cells were transfected with the indicated amounts of pcDNA3.1-hcGAS vector using PEI transfection reagent. Twenty four hours later the cells were harvested by centrifugation and washed once with ice-cold PBS. Cell lysates were prepared using the 2'3'-cGAMP EIA protocol and 2'3'-cGAMP was quantified according to the manufacturer's instructions (Arbor Assays).

**Western blotting.** HEK293T cells were plated at a density of 750,000 cells per well of a 6-well cell culture plate. The next day cells were transfected with 1 µg pcDNA3.1-hcGAS or empty vector using PEI transfection reagent. Twenty four hours later cells were harvested by centrifugation, and the cell pellets were lysed in Pierce RIPA buffer (Thermo Scientific) supplemented with Halt Protease and Phosphatase Inhibitor Cocktail (Thermo Scientific). Lysates were clarified by centrifugation, and protein content was normalized using Pierce BCA Protein Assay Kit (Thermo Scientific). In total, 30 µg of protein per condition were loaded onto Any kD Mini-PROTEAN TGX Precast Protein Gels (Bio-Rad) and separated by SDS-PAGE. Proteins were then transferred onto nitrocellulose membranes (Bio-Rad) at 100 V for 90 min at 4 °C. The membranes were then air dried for one hour and blocked in 5% Blotto, non-fat milk (Santa Cruz Biotechnology) dissolved in 1× TBS for one hour. Membranes were probed overnight in 5% Bovine Serum Albumin (Fisher Scientific) dissolved in 1 X TBS-T with anti-cGAS Rabbit mAb (1:1000) and anti-β-Actin Mouse mAb (1:1000) (Cell Signaling Technology). Proteins were visualized using IRDye 800CW Goat anti-Rabbit IgG Secondary Antibody (1:10000) and IRDye 680RD Goat anti-Mouse IgG Secondary Antibody (1:10000) (LI-COR Biosciences). All wash steps were carried out using 1× TBS-T. Blots were imaged using an Odyssey Fc System (LI-COR Biosciences). Rabbit anti-cGAS (D1D3G; cat. no. 15102) and mouse anti-β-Actin (8H10D10; cat. no. 3700) monoclonal antibodies were obtained from Cell Signaling Technology.

**Reporting summary.** Further information on research design is available in the Nature Research Reporting Summary linked to this article.

## Data availability

X-ray crystallographic structure files of human STING CTD were obtained from the Protein Data Bank (PDB) using accession codes: 4F5D [https://doi.org/10.2210/pdb4F5D/pdb] and 4F5E [https://doi.org/10.2210/pdb4f5e/pdb]. The datasets generated during and/or analyzed during the current study are available from the corresponding author on reasonable request.

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

## Acknowledgements

We thank all of the members of the Woodward lab for helpful discussions. This research was supported by the Cell Analysis Facility Flow Cytometry and Imaging Core in the Department of Immunology at the University of Washington. Calf-thymus (CT) DNA and Interferon Stimulatory DNA (ISD) were a kind gift from Dr. Daniel Stetson. Poly(ethyleneimine) (PEI) was a kind gift from Dr. Keith Elkon. pSLIK-Puromycin lentiviral vector was a kind gift from Dr. Andrew Oberst. pcDNA3.1 plasmid encoding full-length human cGAS and STING were kind gifts from Dr. Genhong Cheng. pcDNA4 plasmids encoding *B. subtilis* DisA, *P. aeruginosa* WspR*, Vaccinia Virus (VacV) Poxin (WT), and Poxin (H17A) were a kind gift from Dr. Philip Kranzusch. pET15b-mKO2-12AA-mTFP was a kind gift from Dr. Samuel Miller. A.J.P is supported in part by a Public Health Service, National Research Service Award, T32GM007270, from the National Institute of General Medical Sciences. S.A. Z. is supported by the Seattle ARCS foundation, as well as grants from the University of Washington/Fred Hutchinson Cancer Research Center Viral Pathogenesis Training Program (2T32AI083203), the University of Washington Medical Scientist Training Program (2T32GM007266), and a Ruth L. Kirschstein Predoctoral Fellowship (1F30CA239659-01A1). This work was supported by National Institutes of Health Grants 1R21AI137758-01 and 1R21AI153820-01. Funding for open access charge: National Institutes of Health Grant 1R21AI153820-01.

## Author contributions

A.J.P. conceived the study, designed and performed experiments, analyzed data, and wrote the manuscript. S.A.Z. designed and performed experiments, analyzed data, and wrote the manuscript J.J.W. conceived the study, obtained funding, and wrote the manuscript.

## Competing interests

The authors declare no competing interests.
