## [Peer Review File · Nature Communications]

Reviewers' Comments:

Reviewer #1:

Remarks to the Author:

Pollock, Zaver, and Woodward developed a genetically-encoded FRET sensor of cyclic dinucleotides based on the mouse STING binding domain. The authors characterize their sensor, BioSTING, with studies using purified protein, lysates, and live HEK cells. They demonstrate applications of BioSTING in several promising in vitro formats for both endpoint and time course experiments, including flow cytometry and imaging applications. Overall, the manuscript is generally well written and the results are clear and support their conclusions for the most part, with a few major and minor comments that need clarification:

Major Points:

1) BioSTING has a high affinity for cGAMP of ~ 50 nM, which the authors state matches physiologically relevant concentrations. There is a major concern that the sensor itself may still interfere with signaling by buffering fluctuations in cGAMP levels similar to the concern faced with calcium sensors. Expression levels of genetically encoded sensors may range from 0.1 to 1 μ M for bright expression. With the sensor's high affinity and slow dissociation kinetics this could bind up cGAS-produced cGAMP after cell activation to alter the kinetics or attenuate the amplitude downstream signaling. The authors need to test that this is not happening, perhaps by IRF dimerization or IFN expression experiments with cells expressing the STING pathway that have been transfected with and without sensor and mutant control sensor.

2) BioSTING is proposed to operate via intramolecular FRET within the one monomer that exhibits a conformational change, while the other monomer does not. However, BioSTING still relies on dimerization for nucleotide binding. Is there any evidence of heterodimerization between a BioSTING monomer and endogenous STING? If this were to occur, sensor expression could again affect downstream signaling. Perhaps the authors could comment if there is any evidence of ER localization of BioSTING in a STING expressing cell type, which might suggest interactions with endogenous STING. Or maybe do a pulldown experiment?

3) For the discussion of Fig. 2 and BioSTING's application to quantification in cell extracts, there is still a concern that other cellular factors could alter the sensor's performance and skew quantification. The authors could compare the sensor's dose responses for purified protein and CDNs in chemically defined solutions (as in Fig. 1) versus doped into untransfected cell lysates.

4) What is the variance in the baseline absolute FRET ratios when BioSTING is expressed in unstimulated cells? A FRET dynamic range of 20% is decent for a first-generation sensor in which the authors have done so much to admirably show that it is useful. However, it poses a concern with quantification of cGAMP concentrations because this relies on a standard curve based on a logarithmic concentration scale that is prone to large errors. From the flow cytometry data in Fig. S2e it looks like the variance is large relative to the maximum FRET change, so it would be appropriate to discuss the need for if not demonstrate a way to calibrate the responses, at least perhaps with a maximal stimulation point.

5) Overall the explanation are very clear, with the one exception being the discussion of intracellular cGAMP measurement in Fig.3 and Fig. S2 and the Discussion. The authors compare empty vector control to cGAS over expression. With cGAS overexpression they estimate micromolar concentrations of cGAMP, but there needs to be more explicit explanation of where estimates of "physiologically relevant" cGAMP concentrations are coming from because heterologous expression of cGAS in HEK cells is the only system used. Similar to point (1) above, at minimum there should be some test with maybe an immune cell line with intact STING signaling, even if with just extracts tested in parallel with BioSTING and EIA quantification.

Minor Points:

- 1) Fig.3's Legend appears to have gotten jumbled a bit with (a) and (c) reversed.
- 2) It would be helpful to give some rationale for the choice of the mTFP-mKO FRET pair in the first results section where the engineering is described. It seems like it was chosen with the intent for flow cytometry applications, but is that true? Were there other reasons that might be of interest for others looking to design FRET sensors?
- 3) Is there any FRET ratio drift when the sensor (or the control mutant) are expressed by itself in unactivated cells? Difference in maturation time could cause drift in basal FRET ratio signals that would through off quantitation for longer time course experiments.

Comment:

Related to point (2) above - Do the authors think that the asymmetry between monomer conformational dynamics is true for a functional BioSTING dimer? If only one monomer is actually undergoing the conformational change, then the other monomer is reducing BioSTING's practical dynamic range because your signal change is sitting on top of a high baseline fluorescence and you would only every expect half of your molecules to exhibit a FRET change. Is there any way to define an asymmetrical tandem dimer in which the dynamic monomer is fused to the FPs and the static monomer is bare?

Reviewer #2:

None

Reviewer #3:

Remarks to the Author:

Detection methods of CDNs are limited, and there is an urgent need for a detection method of real-time monitoring of CDN synthase activity and inhibition. A recent paper by Mardjuki et al reported an assay that can quantify cGAMP in solution and in cell lysates. Compared to the Mardjuki method, BioSTING is the first method that can measure cGAMP concentrations in live cells and, thus, provides a major advance to the field. However, the ability of this tool to quantify cGAMP accurately in vitro has not been demonstrated in an as robust manner as Mardjuki's method. The author should either tune down the language and acknowledge this or perform a few key experiments.

Major comments:

1. Can the authors determine the range of quantification of BioSTING? This is significant because LC-MS/M and the Mardjuki method are robustly quantitative and EIA can be quantitative when cGAMP is diluted into its dynamic range. For example, in Figure 1E - Why is there a large jump in signal between 125 and 250 ng of plasmid? Does this speak to the lack of dynamic range of the sensor (it underreports at low concentration and overreports at higher concentrations) or is this actually reflective of how much intracellular cGAMP there is? Can cGAMP be quantified here using another method? It is important to demonstrate the quantitative powers of BioSTING or the author can remove such claims.
2. The authors note that the FRET maximum is 20% $\Delta R/R_0$. Can the authors clearly state the sensor's range, max and min in terms of $\Delta R/R_0$ and the corresponding cGAMP concentration?
3. The IC50 is shifted higher in Fig 3d (nucleofecting cGAMP into cells) compared to Fig 1d (in vitro). Is this a short coming of BioSTING or a limitation of nucleofection?
4. Line 230. Do authors always have to measure with EIA/another method in order to quantify, or can you do a standard curve in cells? Figure 3D has no points below 3 μM or so, besides the 0 nM cGAMP point. Would like to see points below this in the standard curve to see how well the standard curve can actually quantify intracellular cGAMP, because without this data it is not valid to then conclude that there is 5-50 nM cGAMP in cells (Line 234).
5. A c-di-GMP BRET sensor was reported by Dippel et al. Is BioSTING better in some way?
6. Given that the single mutants Y239S T262A and R231H don't appear to affect 3'3'-c-di-GMP binding, it's surprising that the triple mutant has reduced FRET signal. As such, it is unclear if the

triple mutant is a weaker 3'3'-c-di-GMP binder or if the reduced FRET signal is unrelated to 3'3'-c-di-GMP binding. Can the authors test the triple mutant's affinity for 3'3'-c-di-GMP in vitro? Alternatively, can they show that the triple mutant is capable of reaching the same max FRET response as WT BioSTING?

7. The authors have used HEK 293Ts as a model cell line, which do not express cGAS or STING endogenously. Can they demonstrate use of BioSTING in cell lines that expresses endogenous levels of cGAS (and stimulate with DNA)? In addition, can they demonstrate use in a cell line that already expresses STING? Will competition between endogenous STING and BioSTING for cGAMP affect the detection of cGAMP by BioSTING?

8. Does expression level of BioSTING affect FRET signal for experiments that involve transfected/electroporated/extracellular cGAMP? Given the slow dissociation rate of cGAMP from STING demonstrated in Figure 1g, it's likely that BioSTING sequesters cGAMP, and that higher BioSTING levels will result in artificially higher intracellular cGAMP levels. The authors should perform a doxycycline titration in their BioSTING HEK293T cells and evaluate how that affects FRET signal in response to a constant concentration of extracellular cGAMP.

9. Figure 5b should be accompanied by a Western blot to show that the levels of expressed Poxin are indeed constant across the cGAS titration (minor point).

Typos:

1. Line 111: STING isoforms should be referred to as STING alleles. (isoforms typically refer to different proteins originating from the same gene as a result of alternative splicing).

2. Line 117: double that (that, ..., that).

3. Line 214: Should be 2'3'-cGAMP, not 2',3'-cGAMP.

4. Figure 1E: Please add what concentration of nucleotide was used to the caption.

5. Figure 1F: Please add how much excess nucleotide is present.

6. Figure 1G: Please add how long was the interval between the addition of hot cGAMP was the addition of cold cGAMP. What concentrations of CDNs were used?

7. Figure S2D: Assuming 100% efficiency of transfection, what concentrations of cGAMP would end up being in the cells. The amount of cGAMP in ng is not helpful as a reference point, and the x-axis in 3D is given as concentrations of cGAMP.

8. Figure 4E: Please add how much of each plasmids was transfected.

9. Figure legends for 3A and 3C are mixed up.

10. Figure legends for 3G and 3H are mixed up.

Response to Referees:

Reviewer #1 (Remarks to the Author):

Pollock, Zaver, and Woodward developed a genetically-encoded FRET sensor of cyclic dinucleotides based on the mouse STING binding domain. The authors characterize their sensor, BioSTING, with studies using purified protein, lysates, and live HEK cells. They demonstrate applications of BioSTING in several promising *in vitro* formats for both endpoint and time course experiments, including flow cytometry and imaging applications. Overall, the manuscript is generally well written and the results are clear and support their conclusions for the most part, with a few major and minor comments that need clarification:

We thank the reviewer for their interest in our paper and for their constructive feedback to improve our manuscript. We have addressed the reviewer's comments below.

Major Points:

1) BioSTING has a high affinity for cGAMP of ~50nM, which the authors state matches physiologically relevant concentrations. There is a major concern that the sensor itself may still interfere with signaling by buffering fluctuations in cGAMP levels similar to the concern faced with calcium sensors. Expression levels of genetically encoded sensors may range from 0.1 to 1 μ M for bright expression. With the sensor's high affinity and slow dissociation kinetics this could bind up cGAS-produced cGAMP after cell activation to alter the kinetics or attenuate the amplitude downstream signaling. The authors need to test that this is not happening, perhaps by IRF dimerization or IFN expression experiments with cells expressing the STING pathway that have been transfected with and without sensor and mutant control sensor.

We were also interested in determining whether BioSTING measurements could be coupled with STING and IFN activity measurements from the same sample. Because of the way we engineered the biosensor, there was a possibility that BioSTING could bind to endogenous STING, thereby altering both FRET responsiveness and STING activation. To test this, we transfected HEK293T cells stably expressing BioSTING with a hSTING expression plasmid. Expression of human STING in this context decreased BioSTING FRET response in the presence and absence of cGAS (Figure 5g). Observation of a FRET decrease upon expression of STING, even in the absence of cGAS, suggests that heterodimerization leads to an altered conformation and that cells must be deficient in STING expression to attain interpretable FRET responses. Although not tested, we expect that, for the same reasons, endogenous STING activation would be impaired by BioSTING. In addition, as the reviewer suggested, it is also possible that endogenous STING can compete with BioSTING for 2'3'-cGAMP (and vice-versa) resulting in a lower FRET signal. This is a current limitation of our first generation biosensor, and we have now included this information in the revised manuscript (lines: 369-378 and 469-477 figure: 5g)

Nevertheless, we would like to emphasize that BioSTING has broad utility in many contexts and there are several ways in which this limitation can be overcome including using cell lines that don't express STING or by making STING KO cell lines. With next generation BioSTING sensors, we hope to make a few improvements including (i) re-engineering the dimerization domain so that this no longer is an issue, (ii) developing BioSTING variants with altered nucleotide specificities, and (iii) testing other assay platforms for use *in vivo*, including BRET. These are of considerable interest to us and these studies provide a solid foundation upon

which to build these second generation tools, but we believe that developing these sensors is out of the scope of the current work.

Due to the limitations of measuring effects downstream of STING as described above, we sought to determine if BioSTING expression levels caused differences in FRET response. We addressed this by taking the population of cells analyzed and splitting them into the top 50% of cells and bottom 50% of cells for multiple experiments including AMAXA, extracellular uptake, and cGAS stimulation by CT-DNA. No significant differences are observed between these two populations. Although this analysis isn't perfect, it suggests that the concentration of BioSTING is not radically altering FRET response.

Reviewer Figure 1:
cGAMP Nucleofection (Figure 3d):

Extracellular cGAMP (Figure 5c):

CT-DNA transfection (Figure 5a):

2) BioSTING is proposed to operate via intramolecular FRET within the one monomer that exhibits a conformational change, while the other monomer does not. However, BioSTING still relies on dimerization for nucleotide binding. Is there any evidence of heterodimerization between a BioSTING monomer and endogenous STING? If this were to occur, sensor expression could again affect downstream signaling. Perhaps the authors could comment if there is any evidence of ER localization of BioSTING in a STING expressing cell type, which might suggest interactions with endogenous STING. Or maybe do a pulldown experiment?

Please see response to point 1).

3) For the discussion of Fig. 2 and BioSTING's application to quantification in cell extracts, there is still a concern that other cellular factors could alter the sensor's performance and skew quantification. The authors could compare the sensor's dose responses for purified protein and CDNs in chemically defined solutions (as in Fig. 1) versus doped into untransfected cell lysates.

For these *in vitro* assays, cellular extracts were prepared by methanol extraction, following the protocol used for preparing samples for LC/MS analysis. Methanol extracts were then dried under vacuum and resuspended in assay buffer. This method also serves to remove most cellular factors like proteins and nucleic acids that could complicate the measurements. This is the same protocol followed by other groups to quantify c-di-GMP from cellular extracts using a protein biosensor [1]. Because BioSTING specifically binds to CDNs and not any other nucleotide species, we have not observed any interference under these assay conditions. For other lysis methods, BioSTING can, in principle, be used as long as the buffer does not denature BioSTING and as long as the sample is not too dilute, but further optimization would be required. Because this is not a major focus of our work, we have toned down the language with regard to the end point, quantitative capabilities of BioSTING compared to other existing methods [multiple changes between lines: 207-212]. Please see also the response to reviewer #3 comment 1.

4) What is the variance in the baseline absolute FRET ratios when BioSTING is expressed in unstimulated cells? A FRET dynamic range of 20% is decent for a first-generation sensor in which the authors have done so much to admirably show that it is useful. However, it poses a concern with quantification of cGAMP concentrations because this relies on a standard curve based on a logarithmic concentration scale that is prone to large errors. From the flow cytometry

data in Fig. S2e it looks like the variance is large relative to the maximum FRET change, so it would be appropriate to discuss the need for if not demonstrate a way to calibrate the responses, at least perhaps with a maximal stimulation point.

For all of the flow cytometry data presented in our manuscript, we have calculated the mean fluorescence intensities (MFI) for the population of more than 10 thousand cells in each sample. We are therefore reporting the average fret ratio ($\Delta R/R_0$) of the entire population for each sample. In essence, this is no different than calculating the average cGAMP concentration in a biological sample using EIA or Mass Spec. By this method of analysis, our assay is quite robust and reproducible across biological replicates. In addition to measuring MFIs, our assay also has the added ability to measure FRET responses (i.e. CDN levels) in individual cells, which is what we attempted to show in Figure S2e. When we perform this type of analysis, we see that CT-DNA transfection causes the entire population to shift towards higher FRET ratios but there is heterogeneity of cGAMP levels in individual cells within this population. We believe that this is due to heterogeneity of cGAMP production and/or heterogeneity of CT-DNA delivery. To address the second point brought up by the reviewer, it is indeed very important to calibrate the assay each time using a fresh standard curve for *in vitro* assays (just as with EIA or mass spectrometry) or empty vector and cGAS overexpression to establish the lower and upper bounds for the experiment. We have highlighted this in the methods section of the manuscript [Lines 621-622 and 649-651].

5) Overall the explanations are very clear, with the one exception being the discussion of intracellular cGAMP measurement in Fig.3 and Fig. S2 and the Discussion. The authors compare empty vector control to cGAS over expression. With cGAS overexpression they estimate micromolar concentrations of cGAMP, but there needs to be more explicit explanation of where estimates of “physiologically relevant” cGAMP concentrations are coming from because heterologous expression of cGAS in HEK cells is the only system used. Similar to point (1) above, at minimum there should be some test with maybe an immune cell line with intact STING signaling, even if with just extracts tested in parallel with BioSTING and EIA quantification.

We thank the reviewer for pointing this out, we have revised the text in the manuscript to make this section more clear (lines: 218-224). Based on literature precedent, physiological cGAMP concentrations, including in immune cell lines, have been shown to range from the low nanomolar to high micromolar range [2]. Titration of cGAS in 293T cells resulted in intracellular 2'3'-cGAMP concentrations that ranged from 5 nM to ~8 μ M based on EIA measurements (Fig 3e-f), and we were able to observe BioSTING FRET responses at all of these concentrations in cells with an EC50 between 5-50 nM. Thus, for intracellular and *in vitro* cGAMP measurements, BioSTING can be employed in any context where cGAMP levels are in the low nanomolar to high micromolar range. For picomolar detection of cGAMP, other methods need to be employed i.e. EIA.

Minor Points:

1) Fig.3's Legend appears to have gotten jumbled a bit with (a) and (c) reversed.

We thank the reviewer for identifying this error. It has now been corrected in the resubmitted manuscript.

2) It would be helpful to give some rationale for the choice of the mTFP-mKO FRET pair in the first results section where the engineering is described. It seems like it was chosen with the

intent for flow cytometry applications, but is that true? Were there other reasons that might be of interest for others looking to design FRET sensors?

We thank the reviewer for highlighting this concept. We also tried to use the eCFP and eYFP FRET pair and obtained no FRET response. We hypothesize that this is due to altered chromophore orientation of the GFP derived eYFP versus the coral derived mKO2 fluor. We followed up on the mTFP/mKO2 FRET pair both because it provided a response we could optimize and because the mTFP/mKO2 FRET pair is brighter and more stable [Line 122-124].

3) Is there any FRET ratio drift when the sensor (or the control mutant) are expressed by itself in unactivated cells? Difference in maturation time could cause drift in basal FRET ration signals that would through off quantitation for longer time course experiments.

We have not observed any difference in BioSTING responses between 16-28 hour inductions, but we have not tested longer inductions. We have tested the same sample by flow cytometry 2 hours apart and observed no change in FRET ratio. We have not tested samples over longer periods of time. Because cGAMP mediated Interferon activation typically peaks by 2-6 hours, most time course experiments using BioSTING would also likely be on the order of 2-6 hours, so FRET ratio drift should not be a problem for these shorter time course experiments.

Comment:

Related to point (2) above - Do the authors think that the asymmetry between monomer conformational dynamics is true for a functional BioSTING dimer? If only one monomer is actually undergoing the conformational change, then the other monomer is reducing BioSTING's practical dynamic range because your signal change is sitting on top of a high baseline fluorescence and you would only every expect half of your molecules to exhibit a FRET change. Is there any way to define an asymmetrical tandem dimer in which the dynamic monomer is fused to the FPs and the static monomer is bare?

The reviewer brings up an interesting point but this seems technically hard to achieve given the limitation pointed out in comment number 1. For future iterations of BioSTING, we will focus on re-engineering the dimerization domain of BioSTING to force homodimerization.

Reviewer #3 (Remarks to the Author):

Detection methods of CDNs are limited, and there is an urgent need for a detection method of real-time monitoring of CDN synthase activity and inhibition. A recent paper by Mardjuki et al reported an assay that can quantify cGAMP in solution and in cell lysates. Compared to the Mardjuki method, BioSTING is the first method that can measure cGAMP concentrations in live cells and, thus, provides a major advance to the field. However, the ability of this tool to quantify cGAMP accurately in vitro has not been demonstrated in an as robust manner as Mardjuki's method. The author should either tune down the language and acknowledge this or perform a few key experiments.

We thank the reviewer for their interest in our paper and for their constructive feedback to improve our manuscript. We have toned down the language of the manuscript and addressed the reviewers comments below (lines 218-224).

Major comments:

1. Can the authors determine the range of quantification of BioSTING? This is significant because LC-MS/M and the Mardjuki method are robustly quantitative and EIA can be

quantitative when cGAMP is diluted into its dynamic range. For example, in Figure 1E – Why is there a large jump in signal between 125 and 250 ng of plasmid? Does this speak to the lack of dynamic range of the sensor (it underreports at low concentration and overreports at higher concentrations) or is this actually reflective of how much intracellular cGAMP there is? Can cGAMP be quantified here using another method? It is important to demonstrate the quantitative powers of BioSTING or the author can remove such claims.

To address the first point, we have now included a table summarizing the parameters of BioSTING in the revised manuscript (Table 1) and included a description (line 130-132). With regards to the second question raised by the reviewer, we performed EIA analysis of HEK293T cells transfected with a titration of cGAS expression vector (Figure 3e-f). Based on this analysis, 60-250 ng of cGAS vector resulted in the largest changes in cGAMP concentrations (corresponding to 5-400 nM intracellular cGAMP). These plasmid concentrations were also where we observed the largest changes in BioSTING FRET response in cells (Figure 3e, 4d and 5f), and these results are consistent with the 12-125 nM dynamic range of BioSTING. Thus, the jump in the signal seen in Figure 2e is reflective of the cGAMP concentration in the sample. This can also be explained in part by the dilution factor chosen. For the experiment in Figure 2e, the samples were methanol extracted and resuspended into 40-50 μ L (i.e. 10-25 fold dilution) of assay buffer. As a result, under these dilution conditions, the response was saturated for some of the samples and low for other samples. We re-graphed these results below for the reviewer's convenience:

Reviewer Figure 2:

HEK293T cells were transfected with indicated concentrations of cGAS expression vector. 24 hours later cells were harvested and lysates were assayed for 2'3'-cGAMP production by EIA analysis (left) or methanol extracts were assayed for BioSTING FRET response (right).

Therefore, in terms of *in vitro* cGAMP quantification from cellular extracts, BioSTING faces some of the same limitations as EIA. BioSTING can be quantitative, although it has a small dynamic range, if the sample in question is diluted appropriately. As we discussed, signal saturation can be overcome by diluting the sample serially. Serial dilutions of the sample can then be used to back calculate the concentration in the original sample, similar to EIA. Indeed, using this approach for saturated samples, we were able to calculate an intracellular concentration of cGAMP that was similar to the concentration that we calculated using EIA analysis (Figure 2f and S2d). For low samples, the number of cells used for the experiment can be scaled up, samples can be diluted less, or cGAMP can be enriched through an enrichment step i.e. STING affinity purification. The major advance of BioSTING *in vitro* is kinetic

characterization of cyclase activity, which none of the other methods brought up by the reviewer are capable of performing; however, BioSTING is also suited for endpoint measurements. Because the dynamic range of BioSTING is narrow compared to other detection methods including the new method proposed by Mardjuki *et. al.* [3], we have toned down the language regarding this potential application of BioSTING in the revised manuscript (line: 64 and 207-224). We have also included a citation for this paper, which came out while our manuscript was under consideration.

We would also, however, like to point out that the new method developed by Mardjuki and colleagues is only suitable for endpoint measurements of cGAMP in cell extracts where the cGAMP has been affinity purified using recombinant STING, in order to remove ATP (a substrate of ENPP1) and AMP (a substrate for PAP) as well as to remove cellular ENPP1 inhibitors all of which would interfere with their assay. Moreover, STING affinity purification must be carefully done using the appropriate amounts of recombinant STING to ensure no cGAMP loss from the sample. This is not a problem that our assay faces. Nevertheless, because this is not a major focus of our work we have toned down our claims.

2. The authors note that the FRET maximum is 20% $\Delta R/R_0$. Can the authors clearly state the sensor's range, max and min in terms of $\Delta R/R_0$ and the corresponding cGAMP concentration?

We thank the reviewer for this suggestion. We have summarized the parameters of BioSTING in Table 1 of the revised manuscript.

3. The IC50 is shifted higher in Fig 3d (nucleofecting cGAMP into cells) compared to Fig 1d (in vitro). Is this a short coming of BioSTING or a limitation of nucleofection?

In our hands, this appears to be a shortcoming of cellular transfection methods including nucleofection (1 second nucleofection using AMAXA) and lipofectamine transfection resulting in incomplete delivery of the nucleotide.

4. Line 230. Do authors always have to measure with EIA/another method in order to quantify, or can you do a standard curve in cells? Figure 3D has no points below 3 μM or so, besides the 0 nM cGAMP point. Would like to see points below this in the standard curve to see how well the standard curve can actually quantify intracellular cGAMP, because without this data it is not valid to then conclude that there is 5-50 nM cGAMP in cells (Line 234).

As stated in point 3), in our hands nucleofection and lipofection of CDNs into 293T cells is not efficient and only a small fraction of the total nucleotides are actually transfected into cells. Transfection experiments using purified 2'3'-cGAMP were performed to confirm that the FRET responses observed in cells was indeed due to detection of cGAMP, not for the purposes of generating a standard curve. We have revised the text in the manuscript to make this point more clear. Other labs including Sam Miller's lab (who pioneered the use of FRET biosensors for c-di-GMP sensing) have calculated intracellular concentrations from FRET ratios by interpolating into an *in vitro* standard curve using recombinant FRET biosensor [4]. For quantification purposes, we could similarly calculate the intracellular concentration by interpolating into an *in vitro* standard curve, as others have done, or doing EIA or LC/MS. For the initial characterization of BioSTING in cells, we believed that EIA or Mass Spec based concentration determination would be the most rigorous, which is why we settled on EIA measurements (Figure 3e-f). We have clarified this in the text (line 235-237).

5. A c-di-GMP BRET sensor was reported by Dippel et al. Is BioSTING better in some way?

The c-di-GMP BRET biosensor is an improvement over the CFP/YFP and mTFP/mKO2 c-di-GMP FRET biosensors developed by Sam Miller's lab as well as other chemiluminescent biosensors developed in Ming Hammond's lab using the c-di-GMP binding protein, YcgR [4-6]. These sensors have been incredibly useful for elucidating c-di-GMP regulation in gram negative bacteria. It is difficult to compare these biosensors because they have different purposes. BioSTING is more broadly useful compared to these biosensors in that it allows for the detection of all cyclic dipurine molecules both *in vitro* and in cells, especially 2'3'-cGAMP for which no such sensor currently exists. In addition, the utility of YcgR based biosensors have not yet been demonstrated in eukaryotic cells. The current version of BioSTING cannot be used in animal studies because of the limitations discussed in the manuscript. We hope to design next generation BRET-compatible versions of BioSTING in future studies to enable cGAMP measurements in deep tissues.

6. Given that the single mutants Y239S T262A and R231H don't appear to affect 3'3'-c-di-GMP binding, it's surprising that the triple mutant has reduced FRET signal. As such, it is unclear if the triple mutant is a weaker 3'3'-c-di-GMP binder or if the reduced FRET signal is unrelated to 3'3'-c-di-GMP binding. Can the authors test the triple mutant's affinity for 3'3'-c-di-GMP *in vitro*? Alternatively, can they show that the triple mutant is capable of reaching the same max FRET response as WT BioSTING?

We agree with the reviewer, more work needs to be done to truly determine the effects of R231 mutations on BioSTING FRET response to c-di-GMP. We decided to test these mutants as a starting point because natural mutations at this residue render STING unresponsive to bacterial 3'3' CDNs but responsive to 2'3' cGAMP. In contrast, these same mutations in a BioSTING background yield very different results. Based on our cellular studies, the Y239S/T262A/R231A mutant reached ~50% the maximum FRET response of WT BioSTING for c-di-GMP. Because we have not fully completed our characterization of these mutants and this is not a major focus of the current work, we have decided to move these results to the supplement (Figure S4) in order to highlight some of our screening efforts, but, as suggested by the reviewer, we have toned down our language and claims in the revised manuscript (lines 307-311). Although it is not necessary for the current study, we also believe that mutating Y167, which stabilizes CDN binding through a pi-stacking interaction, to an alanine or valine will likely generate a BioSTING variant that is blind to all CDNs. We have included this in the discussion of the revised manuscript (lines 469-477). This will be an active area of investigation in our lab, which we hope will be the focus of a separate work.

7. The authors have used HEK 293Ts as a model cell line, which do not express cGAS or STING endogenously. Can they demonstrate use of BioSTING in cell lines that express endogenous levels of cGAS (and stimulate with DNA)? In addition, can they demonstrate use in a cell line that already expresses STING? Will competition between endogenous STING and BioSTING for cGAMP affect the detection of cGAMP by BioSTING?

Please see response to Reviewer #1 comment 1).

8. Does expression level of BioSTING affect FRET signal for experiments that involve transfected/electroporated/extracellular cGAMP? Given the slow dissociation rate of cGAMP from STING demonstrated in Figure 1g, it's likely that BioSTING sequesters cGAMP, and that higher BioSTING levels will result in artificially higher intracellular cGAMP levels. The authors

should perform a doxycycline titration in their BioSTING HEK293T cells and evaluate how that affects FRET signal in response to a constant concentration of extracellular cGAMP.

It is indeed possible that BioSTING could sequester cGAMP thereby enhancing the ability of extracellular nucleotides to be pulled into the cell i.e. by LeChatelier's principle. In preliminary experiments where we titrated BioSTING expression, we observed no significant difference in FRET response to intracellular cGAMP produced by cGAS overexpression. We have not tested the effect of BioSTING expression levels on FRET response to extracellular nucleotides. In an attempt to investigate this issue we re-analyzed multiple experiments splitting analyzed cells into high and low BioSTING expressing populations. We did not see significant differences between these two populations suggesting that, as long as BioSTING can be detected by the flow cytometer, the level of BioSTING expression is not severely altering results. For the reviewer's convenience, we have included the data again below. Please also see the response to reviewer #1 comment 1.

Same as Reviewer Figure 1

cGAMP Nucleofection (Figure 3d):

Extracellular cGAMP (Figure 5c):

CT-DNA transfection (Figure 5a):

9. Figure 5b should be accompanied by a Western blot to show that the levels of expressed Poxin are indeed constant across the cGAS titration (minor point).

These plasmids were a kind gift to us from Philip Kranzusch and were provided to us as untagged expression constructs (described in [7]). Unfortunately, to the best of our knowledge, there are no commercial antibodies against this protein either. We did test for Poxin enzymatic activity in these cells by [32P] labeled CDN hydrolysis assays shown below (Figure S5a-b). As another control experiment, we also co-transfected DisA and WspR* with Poxin, and observed no effect of Poxin on 3'3'-CDN mediated FRET responses, consistent with the 2'3'-cGAMP-selectivity of Poxin (Figure S5c). These data are now included in the manuscript as supplementary figure 5 and described lines: 331-337.

Typos:

We thank the reviewer for identifying these typos and grammatical errors.

1. Line 111: STING isoforms should be referred to as STING alleles. (isoforms typically refer to different proteins originating from the same gene as a result of alternative splicing). We have corrected this error in the resubmitted manuscript.
2. Line 117: double that (that, ..., that). We have corrected this error in the resubmitted manuscript.
3. Line 214: Should be 2'3'-cGAMP, not 2',3'-cGAMP. We have corrected this error in the resubmitted manuscript.
4. Figure 1E: Please add what concentration of nucleotide was used to the caption. We have included this information in the resubmitted manuscript.
5. Figure 1F: Please add how much excess nucleotide is present. We have included this information in the resubmitted manuscript.
6. Figure 1G: Please add how long was the interval between the addition of hot cGAMP was the addition of cold cGAMP. What concentrations of CDNs were used? We have included this information in the resubmitted manuscript.
7. Figure S2D: Assuming 100% efficiency of transfection, what concentrations of cGAMP would end up being in the cells. The amount of cGAMP in ng is not helpful as a reference point, and the x-axis in 3D is given as concentrations of cGAMP. To be consistent with the x-axis in Fig 3d, we have changed the x-axis in S2d to the concentration of 2'3'-cGAMP in the cell culture medium.

8. Figure 4E: Please add how much of each plasmids was transfected. We have included this information in the resubmitted manuscript.
9. Figure legends for 3A and 3C are mixed up. We have corrected this error in the resubmitted manuscript.
10. Figure legends for 3G and 3H are mixed up. We have corrected this error in the resubmitted manuscript.

References:

1. Dippel, A. B., Anderson, W. A., Park, J. H., Yildiz, F. H. & Hammond, M. C. Development of Ratiometric Bioluminescent Sensors for *in Vivo* Detection of Bacterial Signaling. *ACS Chemical Biology* **acschembio.9b00800** (2020) doi:10.1021/acschembio.9b00800.
2. Carozza, J.A. *et al.* Extracellular cGAMP is a cancer-cell-produced immunotransmitter involved in radiation-induced anticancer immunity. *Nature Cancer* **2020**, **1**, 184–196 (2020).
3. Mardjuki, R. E., Carozza, J. A. & Li, L. Development of cGAMP-Luc, a sensitive and precise coupled enzyme assay to measure cGAMP in complex biological samples. *Journal of Biological Chemistry* **295**, 4881–4892 (2020).
4. Mills, E., Petersen, E., Kulasekara, B. R. & Miller, S. I. A direct screen for c-di-GMP modulators reveals a *Salmonella* Typhimurium periplasmic L-arginine–sensing pathway. *Science Signaling* **8**, ra57–ra57 (2015).
5. Petersen, E., Mills, E. & Miller, S. I. Cyclic-di-GMP regulation promotes survival of a slow-replicating subpopulation of intracellular *Salmonella* Typhimurium. *Proceedings of the National Academy of Sciences* **116**, 6335–6340 (2019).
6. Dippel, A. B., Anderson, W. A., Evans, R. S., Deutsch, S. & Hammond, M. C. Chemiluminescent Biosensors for Detection of Second Messenger Cyclic di-GMP. *ACS Chemical Biology* **13**, 1872–1879 (2018).
7. Eaglesham, J.B., Pan, Y., Kupper, T.S., Kranzusch, P.J. Viral and metazoan poxins are cGAMP-specific nucleases that restrict cGAS–STING signalling. *Nature* **566**, 259–263 (2019).

Reviewers' Comments:

Reviewer #1:

Remarks to the Author:

The authors have thoroughly responded to every point raised and clarified all the major points of confusion. As a first-generation sensor, BioSTING has been well characterized, the practical advantages and disadvantages are discussed, and the proofs-of-concept are compelling for the applications of interest. There are no additional concerns for this manuscript.

Reviewer #3:

Remarks to the Author:

For comment 8. The authors compared FRET signal in bottom 50% BioSTING expressing cells to top 50% expressing cells. They claim that "We did not see significant differences between these two populations suggesting that, as long as BioSTING can be detected by the flow cytometer, the level of BioSTING expression is not severely altering results." This analysis is not included in the text.

From the authors' analysis, it seems like the expression level does have an effect on FRET signal. While the differences may not be statistically significant, in each of the three cGAMP delivery methods presented (nucleofection, extracellular, and CT-DNA transfection), the FRET signals of the two populations consistently separate. This suggests that for experiments utilizing BioSTING, expression of the reporter should be tightly controlled. At the very least, the authors should include the analysis of the low/high populations in the manuscript and indicate that controlling reporter expression levels is important when comparing FRET signal between samples.

We are happy with how the authors addressed the rest of the comments and this manuscript should be published after the authors make the changes indicated above.

Response to Referees:

We thank both of the reviewers for their enthusiasm for our paper and for their constructive feedback to improve our manuscript.

Reviewer #1 (Remarks to the Author):

The authors have thoroughly responded to every point raised and clarified all the major points of confusion. As a first-generation sensor, BioSTING has been well characterized, the practical advantages and disadvantages are discussed, and the proofs-of-concept are compelling for the applications of interest. There are no additional concerns for this manuscript.

Reviewer #3 (Remarks to the Author):

For comment 8. The authors compared FRET signal in bottom 50% BioSTING expressing cells to top 50% expressing cells. They claim that "We did not see significant differences between these two populations suggesting that, as long as BioSTING can be detected by the flow cytometer, the level of BioSTING expression is not severely altering results." This analysis is not included in the text.

From the authors' analysis, it seems like the expression level does have an effect on FRET signal. While the differences may not be statistically significant, in each of the three cGAMP delivery methods presented (nucleofection, extracellular, and CT-DNA transfection), the FRET signals of the two populations consistently separate. This suggests that for experiments utilizing BioSTING, expression of the reporter should be tightly controlled. At the very least, the authors should include the analysis of the low/high populations in the manuscript and indicate that controlling reporter expression levels is important when comparing FRET signal between samples.

We are happy with how the authors addressed the rest of the comments and this manuscript should be published after the authors make the changes indicated above.

In the revised manuscript, we have now included our analysis of the effect of BioSTING expression levels on FRET responses as Supplementary Figure 4. We have also revised the text to highlight these data (lines: 267-272).